# Air Pollution Patterns Mapping of SO$_2$, NO$_2$, and CO Derived from TROPOMI over Central-East Europe

Beata Wieczorek 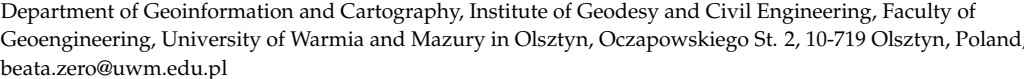

Department of Geoinformation and Cartography, Institute of Geodesy and Civil Engineering, Faculty of Geoengineering, University of Warmia and Mazury in Olsztyn, Oczapowskiego St. 2, 10-719 Olsztyn, Poland; beata.zero@uwm.edu.pl

**Abstract:** The analysis of changes in the level of air pollution concentration allows for the control of air quality and its compliance with the normative requirements. Currently, every country in Europe implements air quality monitoring. However, during emergencies in areas that are often difficult to monitor, the only source of information is geospatial data obtained by means of Earth observation techniques. The aims of this study were to estimate the amounts of pollutant concentrations and develop a pattern of spatiotemporal changes in Central and Eastern Europe in Poland and Ukraine. Due to the ongoing military operations in Ukraine, it is an area that is difficult to access. Pollution from industrial facilities, fires, collapsed buildings, and the use of explosive weapons poses a threat to air quality. Additionally, the impact of war on air pollution concentration levels remains unclear. This work characterized the changes in the distribution of sulfur dioxide, nitrogen dioxide and carbon monoxide concentrations in 2018–2022 in local zones in both countries. Publicly available TROPOMI-S5 satellite data were used for this study, which were compared with measurements from ground stations in Poland. It has been estimated that the concentration of NO$_2$ (+0.67 ± 0.47 μmol/m$^2$) in Poland has increased and the level of SO$_2$ and CO have decreased in both studied areas: in Poland (−161.67 ± 5.48 μmol/m$^2$, −470.85 ± 82.81 μmol/m$^2$) and in Ukraine (−32.56 ± 23.51 μmol/m$^2$, −438.04 ± 80.76 μmol/m$^2$). The concentration of NO$_2$ in Ukraine has decreased by −0.28 ± 0.21 μmol/m$^2$.

**Keywords:** TROPOMI; air pollution; spatiotemporal analysis; Earth observation data; military operations; war in Ukraine

## 1. Introduction

Monitoring and air protection are hindered by the enormous dynamics of the atmosphere, which is the main route for the spread of pollutants and exchange between other elements of the environment [1]. Taking measurements over specific units of time allows for the tracking of changes and their trends and the prediction of the effects. Among the many substances polluting the atmospheric air, a certain number of them are emitted by mostly anthropogenic sources, and these pollutants are treated as characteristic pollutants. These include gaseous pollutants, i.e., gases and vapors of chemical compounds (e.g., carbon oxides (CO and CO$_2$), sulphur (SO$_2$ and SO$_3$), and nitrogen oxides (NOx)); solids, i.e., inorganic and organic particles (PM10, PM2.5); droplet liquids; and biological contamination (microorganisms: viruses, bacteria, and fungi). Due to the way pollutants spread, sources can be divided into point, linear (e.g., communication routes), and surface sources. Each emitter is characterized by technical parameters that determine the spread of pollutants, the most important of which are emission volume (the amount of a substance emitted per unit of time) and the type of emitted pollutants. In addition to the technical parameters of the emitters, the spread of pollutants is affected by meteorological conditions: air temperature, wind direction and speed, and precipitation. These parameters can now be downloaded from the CAMS (Copernicus Atmosphere Monitoring Service), which uses Earth observations in conjunction with numerical models of weather and atmospheric

composition to produce detailed analyses and predictions of atmospheric composition and air quality around the world [2].

The area of Central and Eastern Europe was included in this study. The research on the pattern of changes in spatiotemporal air pollution covers the period 2018–2022. Since 24 February 2022, this region has been experiencing hostilities that are taking place in Ukraine. There are many environmentally sensitive sites in this area of military operations, including industrial and military facilities and nuclear, hydro, and fossil fuel generation sites. The conflict creates the risk of an environmental catastrophe and the quality of atmospheric air may deteriorate due to dust from collapsed buildings or fires [3]. The use of explosive weapons can release many toxic and dangerous chemicals from damaged infrastructure and cause contamination. The pollutants that enter the air and remain in it unchanged are called primary pollutants. Chemical reactions and physical processes occur between atmospheric components and pollutants and, as a result, there are secondary pollutants in the air, which are often more dangerous than primary pollutants [4]. Current observational and theoretical studies are too limited to understand and interpret the environmental impact of the war [5]. Air pollution negatively affects human health not only through direct contact. The failure to protect the flora and fauna from contamination is also a threat. Climate change is considered an ecological disaster, but it is also a major public health problem. Low air quality can lead to a number of respiratory diseases, cardiovascular diseases, and other complications [6,7]. Only a few studies have analyzed the impact of warfare on air pollution [5,8–11]. In the work [10], the authors Zalakeviciute, R., et al. analyzed the air quality in the first weeks of the armed conflict in Ukraine. It was examined on the basis of Sentinel-5P TROPOMI satellite data and ground measurements for PM2.5 concentration. Three two-week periods were adopted for the analysis (from 22 February 2022 to 6 April 2022). The authors of [5] studied the spatial statistics hot spots of $NO_2$ in 2022 in eastern Ukraine. The authors determined that there was a decrease in $NO_2$ concentration by 10.7–27.3% in most Ukrainian cities at the beginning of the war, and war-induced changes in anthropogenic emissions account for about 40% of the reduction in $NO_2$ pollution in major cities, such as Kyiv.

There are many different approaches to the subject of estimating and assessing the degree of air pollution presented in [12–15]. What these studies have in common is the analysis of changes in the concentration distribution on a global and regional scale and the impact on the atmosphere, as well as the validation of air quality models [16–22]. Many studies are concerned with the impact of natural sources such as volcanoes, fires, dust storms, the extraction of gases from the ground, swamps, etc. [23–29]. However, the main causes of atmospheric pollution are artificial sources, primarily human activity [30–32]. Anthropogenic sources of pollution include transport, agriculture, industry, municipal economy, and fuel extraction and combustion processes [33–40]. There are many reasons for exceeding the criterion values of pollutant concentrations, which are taken into account in the assessment of air quality. Recently, many studies have focused on the impact of emissions related to vehicle traffic and emissions from industrial companies during the COVID-19 pandemic and lockdown periods [41–47].

Currently, Earth Observation (EO) systems, together with innovative solutions for data processing and analysis, make it possible to monitor spatial-temporal changes and assess the impact of anthropogenic factors on the environment. Sensors on various platforms (e.g., EOS Aura-OMI, MetOp, Tropomi-S5, EOS-Terra, ENVISAT-SCIAMACHY) provide detailed information (higher spectral, spatial, and temporal resolutions) and more parameters. Currently, observations, both from satellites and from the ground, provide information about air quality.

The aim of this article is to present research on the variability of air pollution and to analyze trends in the territorial zones in Poland and Ukraine for various time scales on the basis of satellite measurements. This study estimated the surface concentrations of $SO_2$ (sulfur dioxide), $NO_2$ (nitrogen dioxide), and CO (carbon monoxide) gases from TROPOMI/Sentinel-5. From 1 November 2018 to 31 October 2022, air above Central and

Eastern Europe was analyzed. Due to the hostilities that have been taking place in Ukraine since 24 February 2022, this study assessed how the war affected changes in emissions of $SO_2$, $NO_2$, and CO in the atmosphere. While many air pollution studies cover the analysis of the entire area, this study was based on zonal characteristics of gas concentrations.

It is also worth mentioning that the study of trends in the area of Ukraine was not verified with ground measurements because the measuring stations have stopped working. In Poland, the assessment of the level of substances in the air, due to the protection of human health, is carried out in zones throughout the country from ~200 monitoring stations. Therefore, the measurements of gas concentrations of $NO_2$ and $SO_2$ from ground-based monitoring stations in Poland were verified with TROPOMI satellite measurements.

## 2. Materials and Methods

### 2.1. Study Site

The research object was the area of Central and Eastern Europe, covering two countries: Poland and Ukraine. Poland is situated between the Baltic Sea in the north and the Sudetes and Carpathians in the south. To the west, it borders Germany, and to the east, it borders Russia, Lithuania, Belarus, and Ukraine, where an armed conflict is currently taking place. The Republic of Ukraine borders the Black Sea and the Sea of Azov, and the Russian Federation to the west (Figure 1). Today, most of the hostilities take place in the industrial area of western Ukraine, often in urban areas. Thus, there is a probability of air pollution with many substances. Substances released into the air persist in the atmosphere for several hours to even weeks. In Poland, the biggest source of air pollution is transport and industry [4].

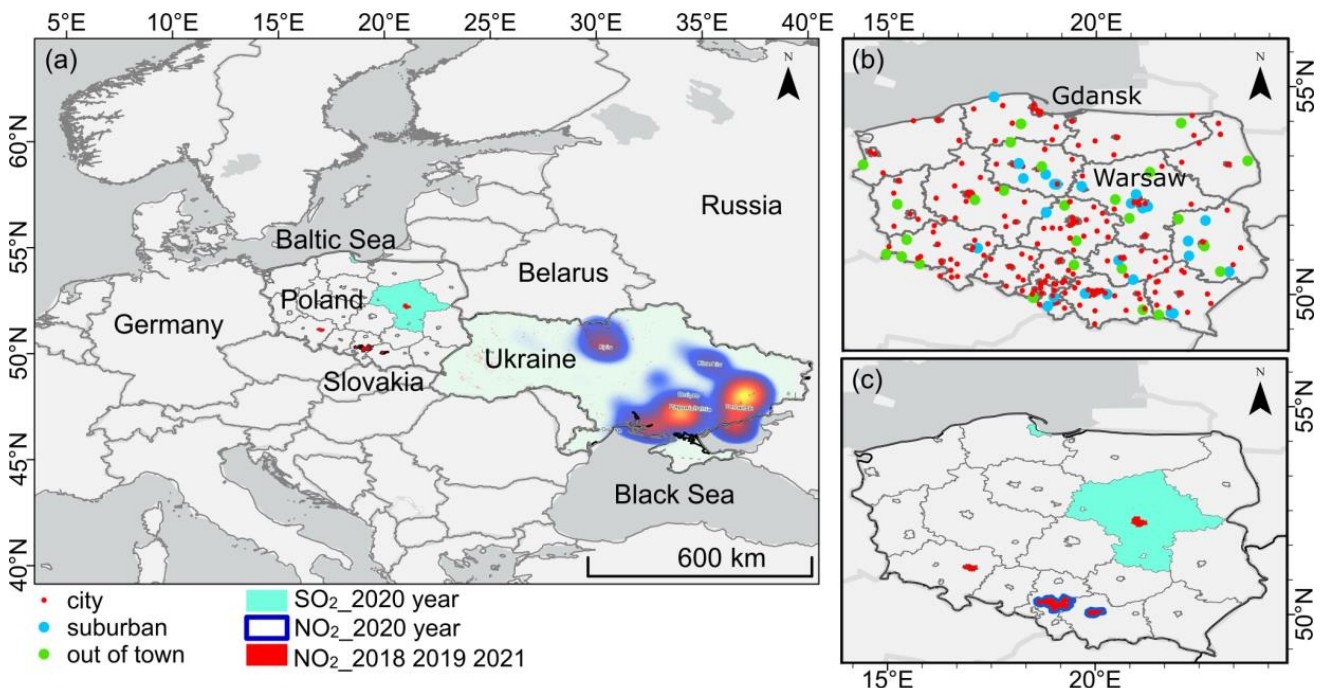

**Figure 1.** The area of Central and Eastern Europe. (**a**) In Poland, areas with increased concentrations of gases are marked; in Ukraine, places of armed conflict are marked (31 October 2022); (**b**) the distribution of ground measuring stations and the ranges of air quality monitoring zones; (**c**) in Poland, areas with high concentrations of $NO_2$ and $SO_2$ are marked (measurements took place in 2018–2021, ground stations).

Ground measurements which exceed the criterion values of gas concentrations were observed in large cities in the south of Poland. Metallurgy, mining, and other branches of heavy industry are developed in this area. It is a densely urbanized area with many roads.

*2.2. Datasets*

2.2.1. Data in Ukraine

The study lacked data from ground measuring stations in Ukraine. As most of the measurement stations have been damaged, there is currently no station included in the European Environment Agency under the European Air Quality Index (https://www.eea.europa.eu, accessed on 30 November 2022). Data on environmental incidents were collected by international NGOs Zoï Environment Network (Switzerland), the OSCE Project Coordinator in Ukraine, and the UN Environment Program and presented on the Ecodozor map (https://ecodozor.org, accessed on 30 November 2022).

Information about events in the locations of industrial facilities and critical infrastructure were obtained from various sources, including authorities, universities, and civil society. Events were also identified using remote sensing data. On the basis of the collected data, an expert assessment was carried out, as a result of which a categorization of environmental threats was created. Risk levels were assigned, taking into account the type of industrial or other activity, the nature and location of events, and other factors. Verified data were presented in the Ecodozor database as reports on the status of various pollutants. The second source of information on pollution was the Purpleair website (https://map.purpleair.com, accessed on 30 November 2022). Users with air pollution sensors uploaded data on PM2.5 concentration levels to the cloud, and these data were presented on the map in real time. The availability of the described data appeared to be insufficient for this study.

2.2.2. Data in Poland

In Poland, the Chief Inspectorate of Environment Protection (CIEP—Główny Inspektorat Ochrony Środowiska (GIOŚ)) is responsible for air quality monitoring. The assessment is based on criteria established to protect health concerns with regards to 12 substances, including $SO_2$, $NO_2$, CO, $C_6H_6$, $O_3$, PM10, PM2.5, and (Pb, As, Cd, Ni, B(a)P) in PM10. Air quality measurements include: agglomerations, zones—cities, and zones—other areas of voivodeships not included in agglomerations and cities (Figure 1). Sulfur dioxide ($SO_2$), nitrogen dioxide ($NO_2$), and carbon monoxide (CO) concentration levels from over 200 measuring stations were analyzed [48]. The criteria values, which were the basis for the classification of zones in the annual assessment for individual pollutants, were defined by national regulations [49].

Current assessment methods include: site measurements; indicative measurements (periodic, cyclical); estimation based on the analysis of information on pollutant emissions and their sources; and calculations using mathematical models. Models of pollutant dispersion in the atmosphere were based on data on emissions and meteorological conditions prevailing in the period for which the modeling was performed. The calculations of pollutant concentrations near the ground were conducted using the air quality model GEM-AQ. It was developed on the basis of the GEM (Global Environmental Multiscale)—a numerical weather forecast model created and used by the Canadian Meteorological Center [22]. The GEM-AQ model is used in the European Copernicus service (CAMS_50 Copernicus Atmosphere Monitoring Service—Regional Production) and as part of the European initiative FAIRMODE (Forum for Air Quality Modeling in Europe).

Calculations using the GEM-AQ model and analyses for the assessment of air quality were performed in two stages on a global grid with variable resolution: 2.5 km (0.025°) and, for 30 agglomerations and cities with >100,000 inhabitants, 0.5 km (0.005°).

2.2.3. Surface Observations

Observations of $SO_2$, $NO_2$, and CO concentrations from 1 November 2018 to 31 October 2021 were obtained from CIEP. Observations from measurement stations where the measurement results exceeded the permissible concentration values were used in this study. A summary of the results of high levels of $NO_2$ and $SO_2$ concentrations is presented in Table 1.

**Table 1.** Areas which exceeded the criterion value in the period 2018–2021, the assessment for $SO_2$ and $NO_2$.

| | 2018 | 2019 | 2020 | 2021 |
|---|---|---|---|---|
| $SO_2$ | – | – | – | [1 h] PL2201—350 μg/m$^3$ <br> [24 h] PL1404—125 μg/m$^3$ |
| $NO_2$ [avg/year] | PL0201—46 μg/m$^3$ <br> PL1201—41–61 μg/m$^3$ <br> PL1401—50 μg/m$^3$ <br> PL2404—55 μg/m$^3$ | PL0201—44 μg/m$^3$ <br> PL1201—41–57 μg/m$^3$ <br> PL1401—50 μg/m$^3$ <br> PL2404—54 μg/m$^3$ | PL1201—49 μg/m$^3$ <br> PL2404—47 μg/m$^3$ <br> PL1201—49 μg/m$^3$ <br> PL2404—47 μg/m$^3$ | PL0201—47 μg/m$^3$ <br> PL1201—50 μg/m$^3$ <br> PL1401—43 μg/m$^3$ <br> PL2404—49 μg/m$^3$ |

The air quality index is calculated on the basis of 1 h results from $SO_2$, $NO_2$, and CO concentration measurements. The limit value for CO is 10,000 μg/m$^3$, which refers to the maximum eight-hour average of the moving averages, calculated every hour from the eight one-hour averages during the day. On the other hand, for $SO_2$, poor air quality is 350 μg/m$^3$, and the alarm level is 500 μg/m$^3$; for $NO_2$ these values are 230 μg/m$^3$ and 400 μg/m$^3$, respectively, if the value is present for three consecutive hours at a given measurement point. The EU air quality standard are as follows: for CO—10 mg/m$^3$ (maximum daily 8-h average); for $SO_2$—350 μg/m$^3$ (1 h); for $NO_2$—200 μg/m$^3$ (1 h); and an annual average of 40 μg/m$^3$ [49].

The concentration of the gaseous substance, averaged for one hour, on the surface of the land at the point with coordinates $X_p$, $Y_p$ is calculated according to the formula:

$$S_{x,y} = \frac{E_g}{\pi \bar{u} \sigma_y \sigma_z} - exp\left(-\frac{y^2}{2\sigma_y^2}\right) exp\left(-\frac{H^2}{2\sigma_z^2}\right) \times 1000 \left[\mu g/m^3\right] \tag{1}$$

where $H$ [m] is the effective height of the emitter; $E_g$ [mg/s] is the maximum emission of the gaseous substance; $x$ [m] is the component of the distance of the emitter to the point for which the calculations are made, parallel to the wind direction; $y$ [m] is the component of the distance from the point for which calculations are made, perpendicular to the wind direction; $\sigma_y$ [m] is the coefficient of horizontal atmospheric diffusion; $\sigma_z$ [m] is the coefficient of vertical atmospheric diffusion; and $\bar{u}$ [m/s] is the average wind speed in the layer from the geometric height of the emitter h to an effective height of the emitter [50].

In 2021, an increased concentration of $SO_2$ was recorded around Gdansk in the area of the measuring station in Nowy Port (zone code PL2201). An area of 10 km$^2$ was estimated to be polluted. The contamination was due to an emergency emission from an industrial plant. The second area was in central Poland, in the Mazowieckie zone (PL1404). The estimated size of the area was 9.5 km$^2$, and the increased concentration of $SO_2$ here also resulted from an emergency emission from an industrial plant. In the case of $NO_2$, exceedances of the permissible levels were recorded in all years around the so-called transport stations, which were located directly on roads with heavy traffic and intended for testing the impact of transportation on air quality. These were the agglomerations of Wroclaw (PL0201), Krakow (PL1201), Warsaw (PL1401) and Upper Silesia (PL2404) [51]. In the years 2018–2021, no CO concentrations were exceeded. The region codes are shown in Figure 2.

### 2.3. Satellite Data Processing

The Sentinel-5 Precursor (S5P) mission is a satellite system designed to monitor the state of the atmosphere in the period from 2017 to 2023. The S5P mission is part of the space component of the Global Monitoring of the Environment and Security (GMES/COPERNICUS). It provides information on air quality, climate, and the ozone layer and measurements for ozone, nitrogen dioxide, sulfur dioxide, carbon monoxide, methane, formaldehyde, aerosol index, and clouds [52].

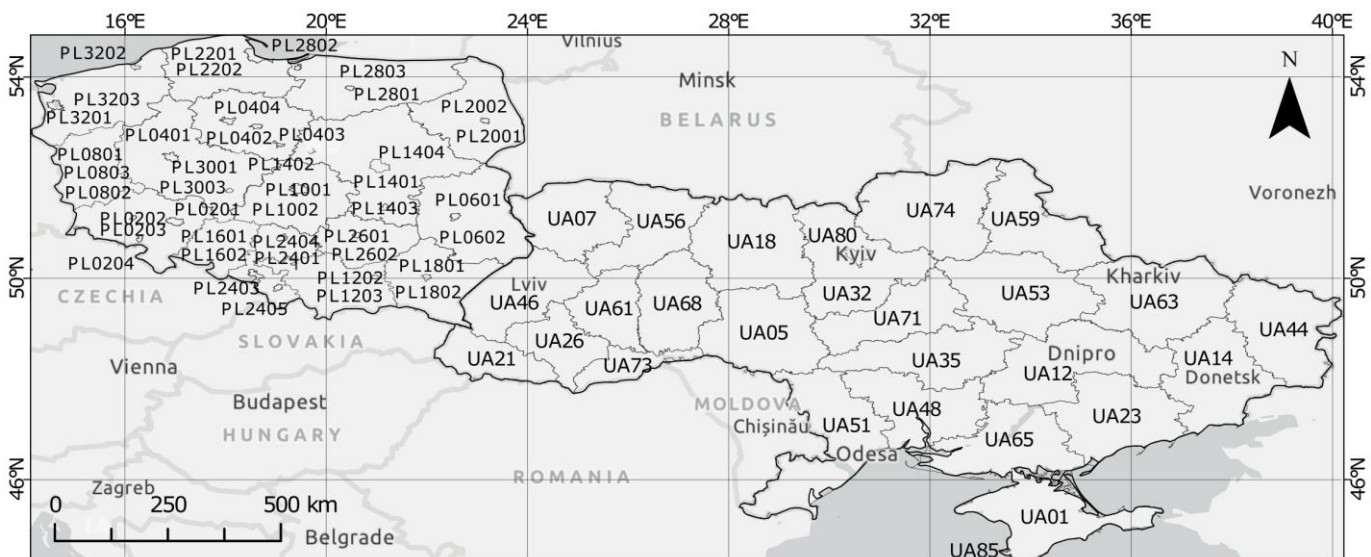

**Figure 2.** Regions for which the spatiotemporal patterns of $SO_2$, $NO_2$, and CO gas concentration levels were studied.

There are four spectrometers on the instrument, imaging in two bands: UV and VIS (270–495 nm); NIR (675–775 nm) and SWIR (2305–2385 nm). The instrument has a spectral resolution of 5–15 km with the ability to reach 50 km at wavelengths <300 nm [52].

The TROPOspheric Monitoring Instrument (TROPOMI) operates in passive remote sensing, imaging swath for approximately 1 s, with spatial sampling of $7 \times 7$ km$^2$ ($7 \times 3.5$ km$^2$; $5.5 \times 3.5$ km$^2$). The S5P TROPOMI sensor collects samples from the Earth's surface with a revisit time of one day. For $NO_2$, $SO_2$, and CO concentration levels, the algorithm provides a vertical column density (VCD) in mol/m$^2$ [53].

One full and, usually, two partial scans are performed over the study area. The author used Level 2 TROPOMI products, and all available data were taken into account. The data were downloaded from 1 November 2018 to 31 October 2022 from the Goddard Earth Sciences Data and Information Services Center (GES DISC). The data were cropped to the area of interest, and they are listed in Table 2.

**Table 2.** Specifications of Sentinel-5 products downloaded from the GES DISC (accessed: 20 August 2020).

| Version, Product | | Temporal Coverage | Spatial Resolution |
|---|---|---|---|
| Sentinel-5P Nitrogen Dioxide Level 2 | | | |
| V1 | S5P_L2__NO$_2$__ | 2018.11.01–2019.08.07 | 7 km $\times$ 3.5 km |
| V1 | S5P_L2__NO$_2$___HiR | 2019.08.06–2021-07-02 | 5.5 km $\times$ 3.5 km |
| V2 | S5P_L2__NO$_2$___HiR | 2021.07.01–2022.10.31 | 5.5 km $\times$ 3.5 km |
| Sentinel-5P TROPOMI Sulphur Dioxide SO$_2$ | | | |
| V1 | S5P_L2__SO$_2$___ | 2018.11.01–2019.08.07 | 7 km $\times$ 3.5 km |
| V1 | S5P_L2__SO$_2$____HiR | 2019.08.06–2020.07.14 | 5.5 km $\times$ 3.5 km |
| V2 | S5P_L2__SO$_2$____HiR | 2020.07.13–2022.10.31 | 5.5 km $\times$ 3.5 km |
| Sentinel-5 Precursor Level 2 Carbon Monoxide | | | |
| V1 | S5P_L2__CO____ | 2018.11.01–2019.08.07 | 7 km $\times$ 7 km |
| V2 | S5P_L2__CO____HiR | 2019.08.06–2021.07.02 | 5.5 km $\times$ 7 km |
| V1 | S5P_L2__CO____HiR | 2021.07.01–2022.10.31 | 5.5 km $\times$ 7 km |

Each set was saved as a multidimensional raster with many variables, such as pollution concentration level, random and systematic error, pressure, cloudiness, quality value for pixels, and others. There was a quality assurance flag, qa_value, available for each pixel.

The quality of the observation depends on many factors that are taken into account when defining the qa_value, so it is a component such as processing errors, presence of clouds or snow/ice, solar radiance, air mass coefficient, any anomalies, and other factors. The "quality assurance value" or qa_value is a continuous variable, ranging from 0 (no output) to 1 (all is well) [53]. In this study, a normalized flag was used as a threshold for rejecting low-quality pixels from usable pixels. The threshold value was established based on processing guidelines; it was 75 for $SO_2$, and 50 for $NO_2$ and CO [54–56].

Pre-processing of TROPOMI data was performed using the ESA Atmospheric Toolbox, and the HARP 1.6 component was used as a library in Python scripts. Level 2 data were processed through conversion to level 3. The conversion was based on a series of parameters that relate simultaneously to dimensions (time, latitude, and longitude), variables (data names and data types), attributes, and coordinates. Data were processed to a grid step of $0.035°$ (grid step) and mean daily rasters were created.

The sulfurdioxide_total_vertical_column measurement, between the surface and the top of the troposphere, was selected for the analysis of $SO_2$ concentration levels.

A dataset was created in which data quality parameters were taken into account. The following parameters are considered in the calculation: qa_value > 0.5; cloud_fraction_crb < 0.3; solar_zenith_angle < 60°; snow_ice_flag < 0.5; sulfurdioxide_total_air_mass_factor_polluted > 0.1; and sulfurdioxide_total_vertical_column > $-0.001$ mol/m$^2$ [56]. For the sets with $SO_2$ in version 2, only qa_value > 0.5 was assumed, because qa_value was adjusted and included the other processing parameters.

Nitrogen dioxide ($NO_2$) and nitrogen oxide (NO) together are usually referred to as nitrogen oxides (NOx = NO + $NO_2$). Tropospheric $NO_2$ acquisition algorithms have been described in the literature [53]. The data file contains the nitrogendioxide_tropospheric_column, which gives the total atmospheric $NO_2$ column between the surface and the top of the troposphere. The data product file gives trace gas concentrations in mol/m$^2$. The processing quality flags contain the individual event that led to processing failure or a precise record of the warnings that occurred during processing. The qa_value indicates whether the trace is covered with clouds or not and whether there is snow or ice on the surface. The recommended pixel filter for qa_value > 0.75 was assumed in this study. It removes cloud-covered scenes (cloud radiance fraction > 0.5), partially snow/ice-covered scenes, errors, and problematic retrievals [54].

The TROPOMI instrument aboard Sentinel-5P also observes the global abundance of CO by exploiting clear sky and cloudy sky Earth radiance measurements. In the 2.3 μm spectral range of the shortwave infrared (SWIR) part of the solar spectrum, TROPOMI clear sky observations provide CO total columns with sensitivity to the tropospheric boundary layer. For cloudy atmospheres, the column sensitivity changes according to the light path (p. 16, [57]). The data file contains the carbonmonoxide_total_column, which gives the total atmospheric column between the surface and the top of the atmosphere (p. 4, [55]). The respective random error originating from the spectral fit is given in the carbonmonoxide_total_column_precision (p. 4, [55]). To avoid the misinterpretation of data quality, only pixels with a qa_value greater than 0.5 were used.

### 2.4. Verification of Satellite Measurement Results with Ground Measurement

The aim of this study was to assess to what extent TROPOMI satellite measurements can be used to determine spatiotemporal patterns in Central and Eastern Europe. In any study, it is important to assess the accuracy of the dataset used. Therefore, the satellite measurements were compared with the available measurements from ground-based monitoring stations in Poland.

The satellite sensor observes the whole integrated tropospheric column, whereas the measurement from the ground station concerns the boundary layer directly affected by the Earth's surface and whether there are significant emissions, e.g., NOx [21,22].

Measurements from CIEP ground stations [48] provide verified information on changes in the levels of air pollution. In this study, 91 stations for $SO_2$ measurements and 115 $NO_2$

stations were selected. The data obtained were in the form of time series of daily averages. Data gaps in the time series were removed by interpolation.

For areas with high gas concentrations, cross-correlation analysis was used to validate the results and statistically explain the correspondence between the measurement from the TROPOMI spectrometric sensor and the corresponding average of ground station measurements.

The level of $SO_2$ emissions was examined at two points where the permissible concentration from ground measurements was exceeded: PmGdaWyzwolenie, Tricity zone (PL2201) and MzBialaKmiciMOB (PL1404)—measurement start date from 1 January 2019. Additionally, a point in the south of Poland near the town of Gorlice was selected for the study, where the average high level of $SO_2$ concentration was recorded from the satellite measurement. The nearest measuring station, MpSzymbaGorl (PL1203), was located 15 km away. Another adopted point was in Wałbrzych in the south of Poland—DsWalbrzWyso (PL0203). The measuring station was located in a place of high concentration.

Measurements from ground stations which exceeded the normative values of $NO_2$ gas concentrations cover the areas of the agglomerations of Wroclaw (PL0201), Krakow (PL1201), Warsaw (PL1401), and Upper Silesia (PL2404).

A preliminary study of the daily values from satellite measurements showed that the data contain measurement gaps. In particular, in the time series of $SO_2$ concentration. Therefore, it was assumed that in the absence of data, in order to fill the time series, a neighboring pixel showing a similar trend could be defined as a reference point to fill in the missing values. Thus, the average monthly measurements of gas concentration were adopted for the analysis. The ranges of the variables in the two measurements were different, so variables with a larger range could have had an undue influence on the results. The data were in different scales from ground stations [$\mu g/m^3$] and satellite data [$mol/m^2$]. Measurements were standardized using the robust method according to Equation (2):

$$x_i = \frac{x - Mdn(x)}{IQR(x)} \qquad (2)$$

where $x_i$ is the standardized value, $x$ is the source value, *Mdn(x)* is the median of the data, and *IQR(x)* is the interquartile range of the data.

Observations from the station were burdened with single values of gas concentration exceedances. The adopted method should be effective when trying to mitigate the impact of outliers on the distribution.

Multidimensional raster sampling was performed at all stations. Time series were created from the extracted cell values at each location. On the basis of the calculated monthly averages, mean errors of monthly observations were determined. Calculations were made for ground and satellite measurements. Then, for each station, the linear relationship between the two variables was checked and maps for $SO_2$ and $NO_2$ were created. Pearson correlation analysis provided information about the strength and direction of the linear relationship between the two variables. The Pearson's correlation coefficient, $r$, can range from $-1$ to 1. The further $r$ is from zero, the stronger the linear relationship between the two variables.

### 2.5. Methodology of Determining the Temporal and Spatial Pattern of Air Pollution

Trending air pollution provides information about likely changes and distribution patterns that is useful for assessing the effects of emission mitigation measures. Changes can occur suddenly, and large fluctuations in ambient air quality are a problem. In addition, climatic conditions particularly affect the transport of pollutants over long distances from large point emitters. Low emissions can increase pollution in the immediate vicinity. Thus, activities such as the armed conflict can affect the air quality in Ukraine. Therefore, this work examines how air quality changes in Poland and Ukraine. After the initial stages of image processing and deriving the pollutant parameters for $SO_2$, $NO_2$, and CO, GIS (Geographical Information System) analysis was used to extract the relevant point data and determine the spatiotemporal distribution of the gas pollution. First, raster mosaics were created.

Then, the averages from the period 1 November 2018–31 October 2022 were calculated from multi-dimensional raster data sets. Time series were created using daily averages from both areas. In the initial stage of this study, the trend analysis method was simple linear regression using the least squares method (OLS—Ordinary Least Squares regression).

### 2.5.1. Trends of Air Pollution Parameters

A step-by-step method was adopted to determine air pollution patterns. Spatial analysis solutions in ArcPro 3.0.3 were used. In the first step, a trend analysis was performed from the time series of the analyzed period for three gases. It was verified whether there were characteristic areas where there was an increase or decrease in gases. The basis for the study were the average monthly values of gas concentrations from the area of Europe. In Poland, zonal monitoring of air quality was carried out. Therefore, the calculations were performed in 45 zones. In Ukraine, however, the calculations were made within the boundaries of 27 administrative districts (Figure 2). Average $SO_2$, $NO_2$, and CO values were calculated for these regions. The list of zone names and their abbreviations is included in Appendix A.

Analyses were carried out to assess the occurrence of trends and trends of changes in the years 2018–2021 for the levels of atmospheric gas concentrations. The non-parametric Mann–Kendall test was used for the assessment, and the slope of the trend was estimated using Sen's estimator. The z-score indicates the existence of a statistically significant trend ($z > 0$ indicates an increasing trend and $z < 0$ indicates a decreasing trend). The Mann–Kendall test analyzes the sign of the difference between successively measured values. The newly measured value is compared to all previously measured values, which gives a total of $n(n-1)/2$ possible data pairs ($n$—number of observations). The statistic ($S$) of the Mann–Kendall test is calculated as:

$$S = \sum_{i=1}^{n-1} \sum_{j=i+1}^{n} sgn(x_j - x_i) \tag{3}$$

Substituting $\theta$ for $x_j - x_i$:

$$sgn(\theta) = \begin{cases} 1 & if\ \theta > 0 \\ 0 & if\ \theta = 0 \\ -1 & if\ \theta < 0 \end{cases} \tag{4}$$

The non-parametric rank correlation coefficient in the Mann–Kendall test is the statistic ($\tau$) given by the formula:

$$\tau = \frac{S}{n(n-1)/2} \tag{5}$$

which takes values in the range $[-1, 1]$.

Changes in the analyzed trend over time can be described by the slope coefficient ($\beta$) expressed with the Sen's estimator calculated with all $i < j$ ($i = 1, 2, \ldots, n-1$ and $j = 2, 3, \ldots, n$):

$$\beta = Mdn\left(\frac{x_j - x_i}{j - i}\right) \tag{6}$$

The assessment of the trend of changes in the determined characteristics of gas concentrations in individual years of the considered period was based on the analysis of the statistical significance of the trend [58,59].

It was assumed that a meaningful change takes place when the significance level is $p \in (0.05 \div 0.10)$, only the tendency to change was assumed to be at the significance level $p \in (0.10 \div 0.25)$, whereas changes at a significance level below $p > 0.25$ were considered insignificant.

2.5.2. Determination of the Pattern of Spatiotemporal Changes

Annual percentage changes in the levels of pollutants by substances were examined. A series of 10-month periods from 2019, 2020, 2021, and 2022 from February 1 to October 31 were adopted. The average annual concentrations in the vertical columns of three pollution criteria ($SO_2$, $NO_2$, and CO) for the regions of Poland and Ukraine were determined. The next step was to determine the weights of the three concentrations. For this purpose, the Multicriteria Decision Making (MCDM) method was used, including the Weighted Sum Model (WSM) method. Then, percentage differences between successive years were estimated.

In the next stage, 10 months from 2022 were selected for analysis. In regions that showed an increase or decrease in gas concentrations, statistically significant points were estimated. For this purpose, two statistical tests were used: global Moran's I and Getis-Ord Gi* [60]. The global Moran's I coefficient is a measure of spatial autocorrelation computed across the entire dataset, which determines whether the dataset is significantly distributed, random, or clustered, and computes the feature's deviation from the mean and cross-products with neighboring features. The Getis-Ord Gi* statistic is used to identify significant spatial clusters of high and low values by calculating the Gi* statistic for each object showing changes in the concentration level [61]. All points, along with their neighboring points, are analyzed in the set. A point with a high value is statistically significant if the adjacent points also show high values. Both statistics were calculated in ArcGIS Pro version 3.0.3.

In the last stage, the areas with the highest sums of gas concentrations were located. The analysis was performed for the year 2022. The average levels of gas concentrations and the speed of changes were taken as the criteria. Weights were assumed as a percentage of gases, based on the assumed percentage of gas in dry unpolluted air in the ground-level troposphere. The same parameters were adopted for the average values and the rate of changes. The analysis was carried out on the basis of indexes from the 1–10 scale.

Atmospheric air up to 80 km above sea level has a constant level of the main components, with the exception of water vapor. In addition, due to the dissociation of molecules, ionization of molecules and atoms, and photochemical reactions occurring as a result of solar and cosmic radiation, its composition changes [50]. Therefore, it was assumed that for the composition of dry atmospheric air at the ground surface, the average contents, % by volume, were $NO_x$ (NO + $NO_2$) $10^{-10}$–$10^{-6}$; $SO_2$~$2 \times 10^{-8}$; and CO~$1.2 \times 10^{-5}$.

## 3. Results

In this study, an analysis of changes in the concentration of three gases was carried out, and the average values of the concentration levels calculated from the years 2018–2022 were determined. To comply with SI unit definitions, the TROPOMI $SO_2$, $NO_2$, and CO product data file was left in [mol/m$^2$]. In the case of CO, total columns [mol/m$^2$] were converted to column volume mixing ratio [ppbv] for visualization.

### 3.1. Correlation Analysis and Verification of Measurements

A comparison of the quality of satellite measurements was carried out with ground-based observations. For this purpose, monthly averages of $SO_2$ concentration levels from 91 stations were calculated. For the TROPOMI data, the pixel value at the station locations was extracted. The time distribution of the gas concentration change is shown in Figure 3.

Both $SO_2$ measurements have a similar pattern and show low values in the summer and high values in the winter. Analysis of the course of the time series shows that they are consistent in the summer months. The differences in the $SO_2$ concentration level measurements are visible in the winter months, especially in 2019 and 2020. These changes may result from cloudiness in the autumn and winter. Qualitative parameters, such as cloud cover, were taken into account in the initial stage of satellite data processing. However, the qa_value quality threshold for $SO_2$ concentration can be increased. The value of 0.5 was assumed in this study.

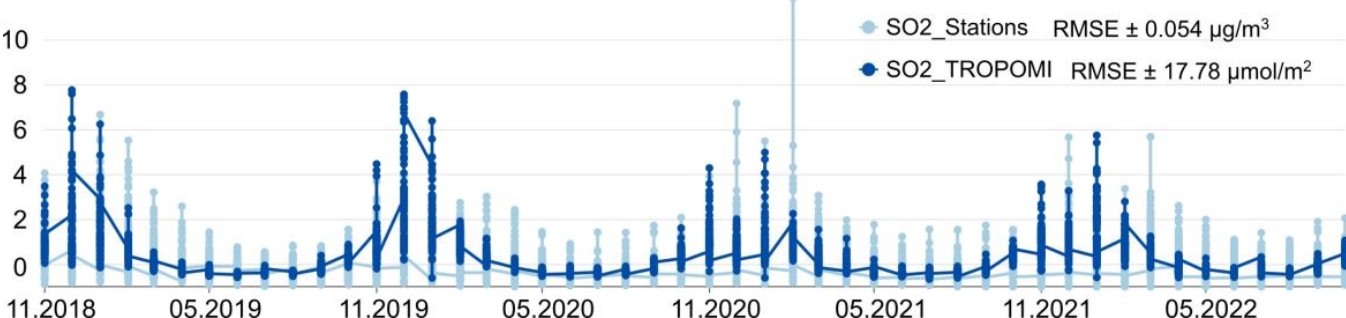

**Figure 3.** Distribution of the average $SO_2$ values, using measurements from ground stations and the pixel value from the TROPOMI measurement at the location of the station (standardized measurements).

The assessment of air quality from satellite data was carried out for monitoring stations with high concentrations of $SO_2$. The Tri-City Agglomeration zone, the Mazowiecka zone, and two zones from the south of Poland were selected. The results of the analysis are compiled in Appendix B, Figure A1.

The time series of $NO_2$ concentration measurements are shown in Figure 4. High $NO_2$ values were observed from ground measurements in 2022. These are measurements from stations located in the west of Poland. On the other hand, the outliers from the satellite measurements were located in the Pomeranian zone and in the Warmian-Masurian zone. In December 2021, the highest concentration from satellite observations was recorded in Zakopane in southern Poland. Analyses included the areas where the normative values of $NO_2$ gas concentrations from ground measurements were exceeded. Stations located in the agglomerations of Wroclaw, Krakow, Warsaw, and Kedzierzyn-Kozle (Upper Silesia) were studied. Due to the proximity of several points within one agglomeration, the raster resolution was increased to $0.02°$. The results of the analysis are compiled in Appendix B, Figure A2.

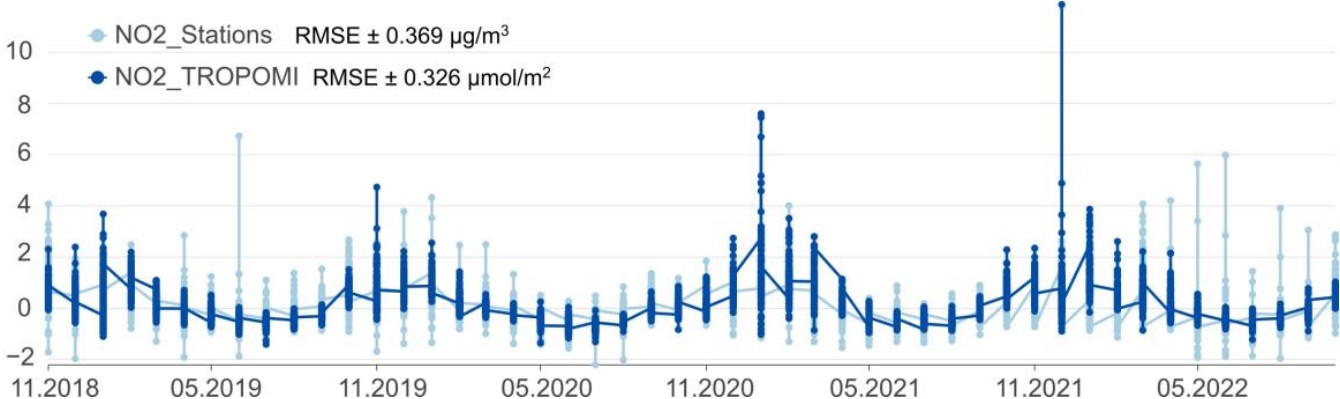

**Figure 4.** Distribution of the average $NO_2$ values, using measurements from ground stations and the pixel value from the TROPOMI measurement at the location of the station (standardized measurements).

The next step was to examine whether there was a statistically significant relationship between the measurements of $SO_2$ and $NO_2$ measured in $[mol/m^2]$ using TROPOMI and the measurements in $[\mu g/m^3]$ from ground monitoring stations. In this work, for the $SO_2$ set, the values of pixels coinciding with the locations of the stations were examined. The Pearson test results showed that the TROPOMI values for some stations were inconsistent with the ground measurement results (Figure 5).

In the tests, it was possible to consider whether the measurements from the stations were collected in automatic or manual mode. In this work, it was checked that the stations for which the correlation coefficient was lower than 0.10 were working in automatic mode. In general, both measurements show a high correlation for $SO_2$ and $NO_2$ concentrations.

Approximately 50% of SO$_2$ measuring stations have a correlation coefficient *r* > 0.50. At some stations, the correlation is strong. The highest value of *r* = 0.79 was obtained by the station in Żywiec, in southern Poland. The highest correlation coefficient of NO$_2$ measurements was *r* = 0.84 at the station in the Silesian zone. In the case of NO$_2$, approximately 78% of measuring stations had a correlation coefficient *r* > 0.50.

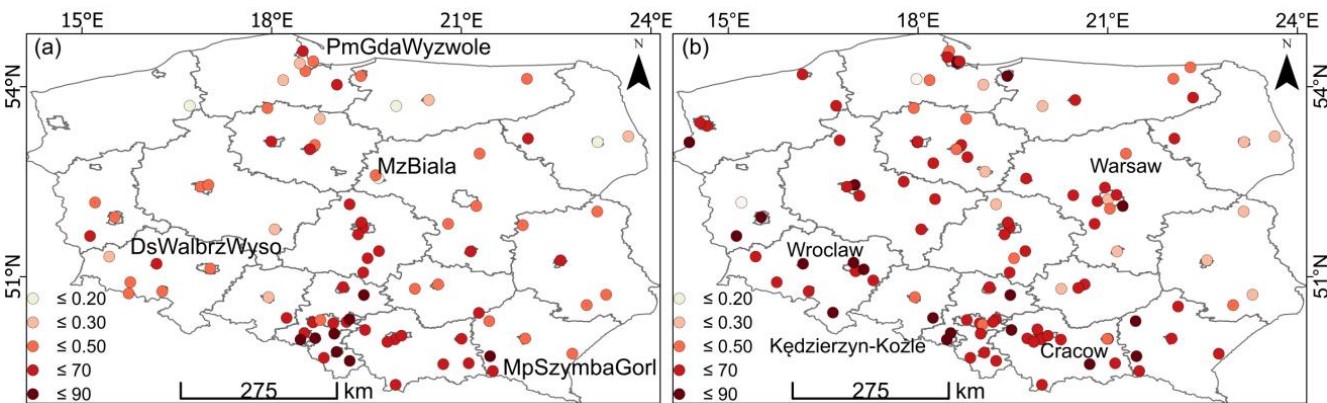

**Figure 5.** The correlation coefficient for linear regression Pearson: (**a**) sulfur dioxide and (**b**) nitrogen dioxide.

### 3.2. Analysis of the Level of Air Pollution in Poland and Ukraine

In the period from 1 November 2018 to 31 October 2022, high concentrations (annual averages) of SO2 were observed in the north of Poland in the area of the Gulf of Gdansk (+1.01 nmol/m$^2$). These measurements coincide with the measurements from 2021 from the ground station. Other areas with a concentration of 0.96 nmol/m$^2$ are the southern part of Poland, the Małopolska region, and occasionally the Lower and Upper Silesian regions. There are two characteristic places in the area of Ukraine's borders: south of Lviv and the western part of the country, near Donetsk (Figure 6). The annual rate of change was −161.67 ± 5.48 μmol/m$^2$ for Poland and −32.56 ± 23.51 μmol/m$^2$ for Ukraine. The value of the trend in the analyzed four-year period in Ukraine decreases slower than that in Poland. In the figure (Figure 7), the days of increased concentration are marked. These are days in January and December.

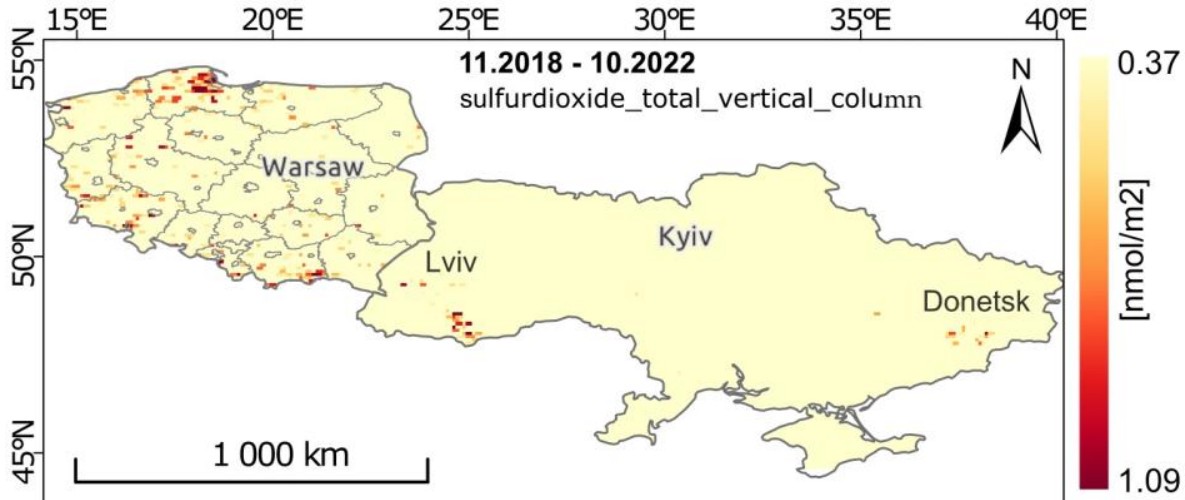

**Figure 6.** Spatial distribution of the average SO$_2$ concentration in Poland and Ukraine in 2018–2022, determined on the basis of the TROPOMI-S5 satellite measurement.

When analyzing the average level of nitrogen dioxide concentration in Poland, there were several intense points. Apart from the area of the central part of the country, they mostly coincide with points from ground measurements (Table 1). In the area of Ukraine, they were primarily the Donetsk region, the Dnieper, the city of Kyiv, and the vicinity of Lviv (Figure 8). The daily concentration of $NO_2$ always increased from June of the current year to June of the following year. The analysis of the speed of average daily values for the study areas showed that the yearly concentration level increased in Poland ($+0.67 \pm 0.47 \ \mu mol/m^2$), whereas in Ukraine it decreased ($-0.28 \pm 0.21 \ \mu mol/m^2$) (Figure 9). The value of the trend in the analyzed four-year period in Ukraine grew faster than in Poland. Higher emissions of carbon monoxide were observed in the area of southern Poland, the region of Donetsk and Dniepr, and the Zakarpattia Oblast south of Lviv (Figure 10). Overall, CO emissions decreased every year in Poland ($-470.85 \pm 82.81 \ \mu mol/m^2$) and in Ukraine ($-438.04 \pm 80.76 \ \mu mol/m^2$). The time series is presented in seven-day intervals (Figure 11), with higher concentrations occurring in the winter season. Higher levels of CO emissions were recorded in areas during August 2021.

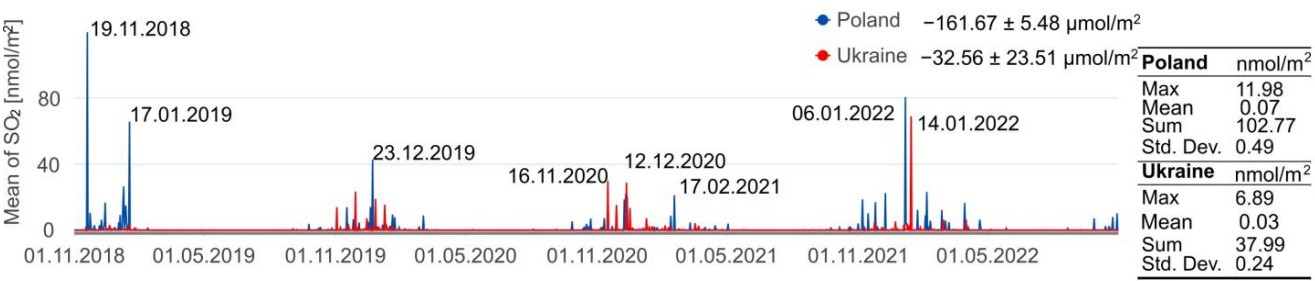

**Figure 7.** Time series of average daily $SO_2$ concentrations in Poland and Ukraine in 2018–2022.

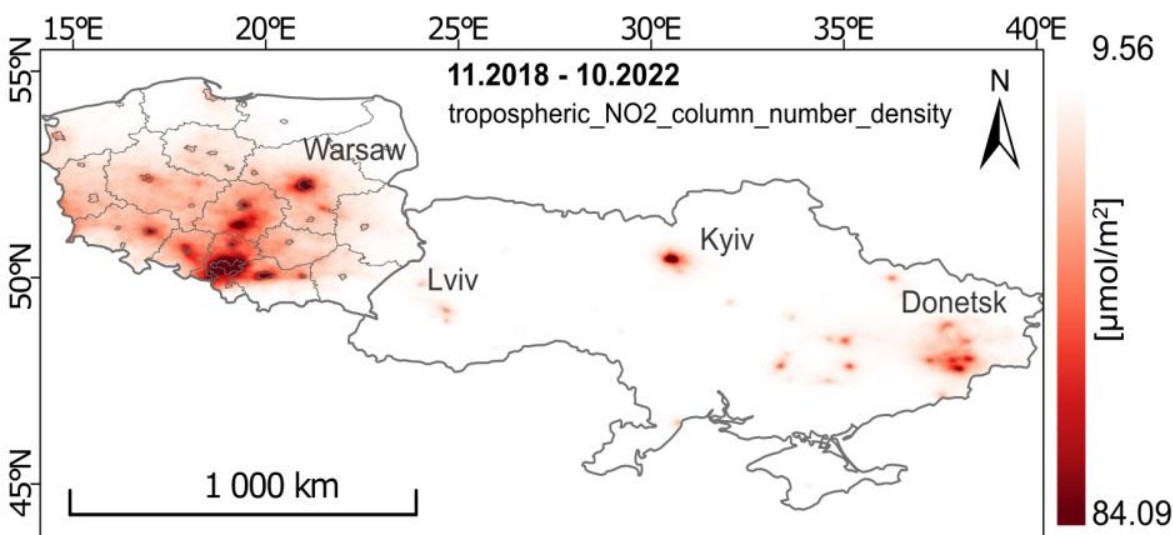

**Figure 8.** Spatial distribution of the average $NO_2$ concentrations in Poland and Ukraine in 2018–2022, determined on the basis of the TROPOMI-S5 satellite measurement.

No significant changes in concentrations were observed in the analysis of time series plots of daily area concentrations of $SO_2$ and $NO_2$.

However, the change in the level of CO concentration in the period from June 2021 to June 2022 needs to be investigated. Therefore, a grid with a resolution of $0.35°$ was created in the area and rasters from June 2021 to May 2022 were selected. A matrix of changes was determined for 172 areas from Poland and 299 from Ukraine. Points were indexed in rows from the top left corner. The concentration distribution was analyzed for large point changes that could affect the area mean. As shown in Figure 12, the increased level

of CO concentration covered the entire territory of Poland and Ukraine. It occurred in August and lasted until September 2021. Grid indices which have values of 0.0412 mol/m² indicate the area of central Poland. In Ukraine, these areas are Kirovohradsk (UA35) and Cherkaska (UA71). The second period of higher CO concentration was in March 2022. This concentration had no spatial distribution. Intensive areas were observed in the northern and south-central parts of Poland. At the same time, it can be seen that in both areas there were regions showing low levels of CO throughout the period. In Figure 12b, the Ivano-Frankivska (UA26), Chernivetska (UA73), Zakarpatska (UA21), and Lvivska (UA46) regions are visible. In Poland, it is primarily the area of the Karkonowski National Park, in the southern part of the Lower Silesian zone (PL0204) (Figure 12a). The analysis of the CO concentration area matrix showed that the reason for the higher concentration from June 2021 to May 2022 was not due to local changes.

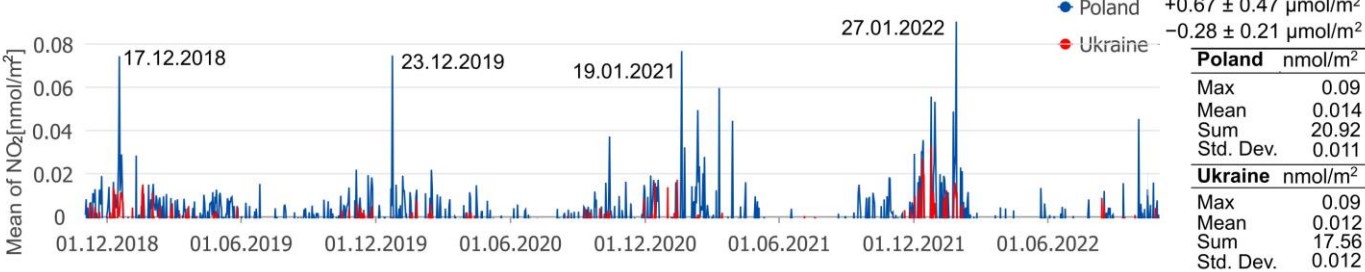

**Figure 9.** Time series of average daily NO₂ concentrations in Poland and Ukraine in 2018–2022.

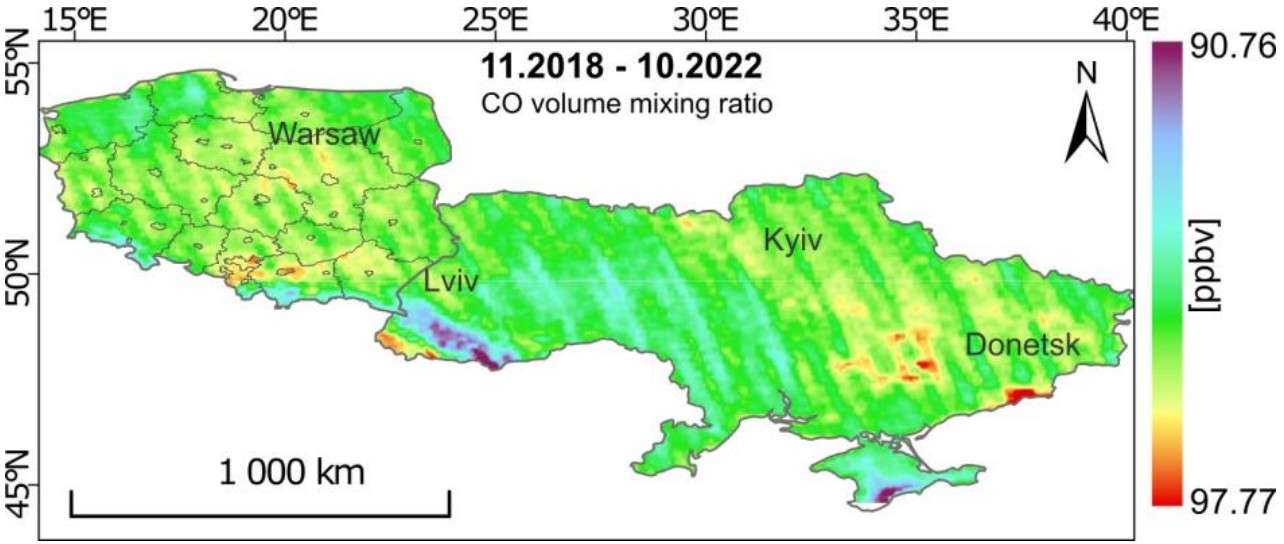

**Figure 10.** Spatial distribution of the average CO concentrations in Poland and Ukraine in 2018–2022, determined on the basis of the TROPOMI-S5 satellite measurement.

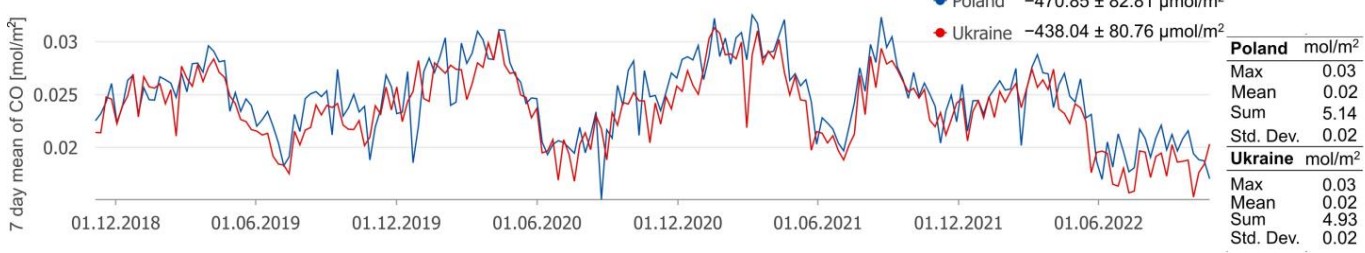

**Figure 11.** Time series of average daily CO concentrations in Poland and Ukraine in 2018–2022.

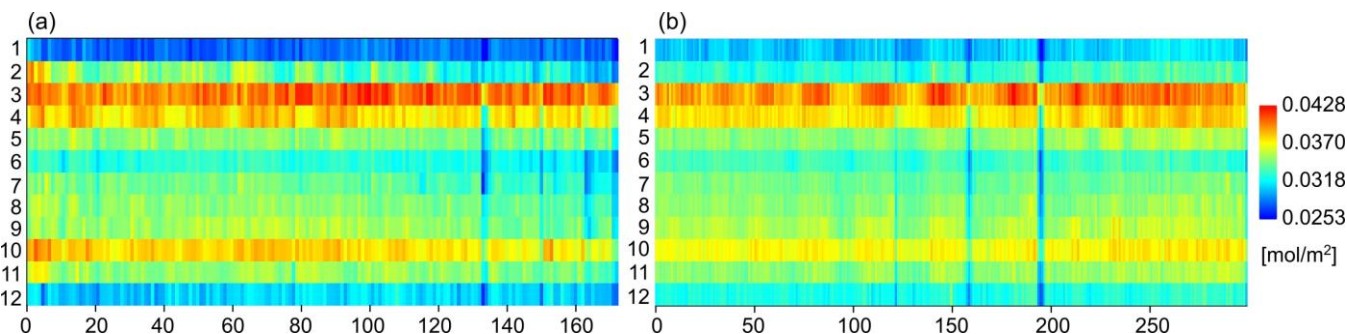

**Figure 12.** Matrix of CO concentration level; the vertical axis shows months from June 2021 to May 2022, the horizontal axis shows the indices of 0.35° grid points; (**a**) the territory of Poland; (**b**) the territory of Ukraine.

### *3.3. Results of Air Pollution Analysis—Mapping*

### 3.3.1. Spatial Distribution

Figure 13 shows comparisons of the four-year average values of $SO_2$, $NO_2$, and CO pollutants in regions in Poland and Ukraine.

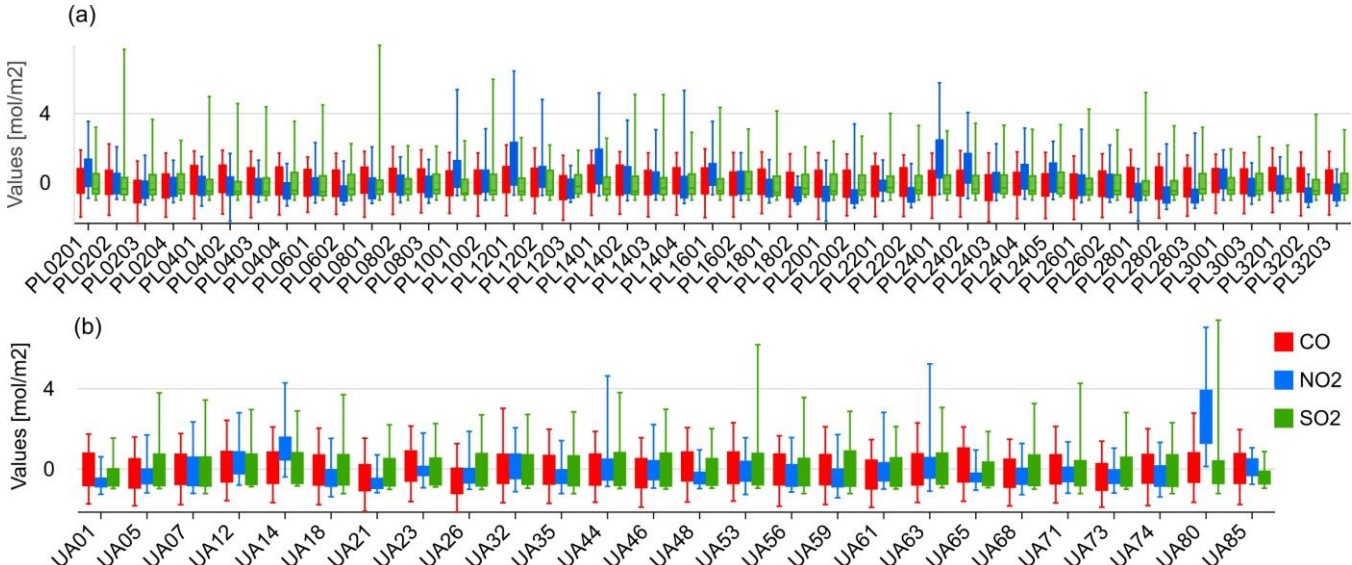

**Figure 13.** Boxplots of distributions of average concentrations of $SO_2$, $NO_2$, and CO pollutants for each zone, from 2018 to 2022; (**a**) for zones in Poland and (**b**) for zones in Ukraine. Boxes cover range from the lower to the upper quartile; the lines and squares inside the boxes represent the median and mean, respectively; and the lower and upper whiskers cover the percentile range from 5% to 95%.

The sets contain $SO_2$ and $NO_2$ outliers. In Poland, these were zones in the western and south-western parts. The highest value of $SO_2$ concentration from monthly satellite data was 10.21 nmol/m$^2$ in the city Gorzów Wlkp. and 9.95 nmol/m$^2$ in the city Legnica. In Ukraine, however, it was 7.10 nmol/m$^2$ in the zone of the city of Kyiv and 6.15 nmol/m$^2$ in the Poltavska region. It can be seen that most of the observations are right-skewed. High concentrations and a large level of dispersion between $NO_2$ values occured in the agglomerations of the largest cities with a high population density in Poland and Ukraine, as well as in areas with developed industries. For example, the Upper Silesian Agglomeration region in Poland and the Donbas region in Ukraine had the highest levels of $NO_2$ pollution, which may be related to highly developed coal mining and processing. The highest value of $NO_2$ concentration from monthly satellite data was 151 μmol/m$^2$ in Warsaw Agglomerations and 104 μmol/m$^2$ in the city of Kyiv. High monthly CO concentration

was recorded in southern Poland (41.09 nmol/m$^2$) and in Ukraine in the Kyivska region (43.32 nmol/m$^2$). On the other hand, high average CO concentrations over a four-year period were recorded in adjacent zones in the south-eastern part of Ukraine (UA65, UA12, UA23). One of the largest power plants in Europe, Ukraine's Zaporizhzhya Nuclear Power Plant, is in this region.

### 3.3.2. Temporal Changes

The average values of the differences in the number of pollutants in the air in individual years in the measurement zones in Poland and administrative districts in Ukraine are shown in Figure 14. It was assumed that the analyses would be carried out in the period from February 1 to October 31 each year. Each bar represents one year with the sum of the gas concentrations. It can be assumed that in 2021, air pollution as a sum of three gases increased in most zones. In 2022, however, it increased only in Pomerania in Poland and in the Warmian-Masurian zone. Considering the growth of each gas separately, there were growth changes, mainly in sulfur dioxide, compared to previous years.

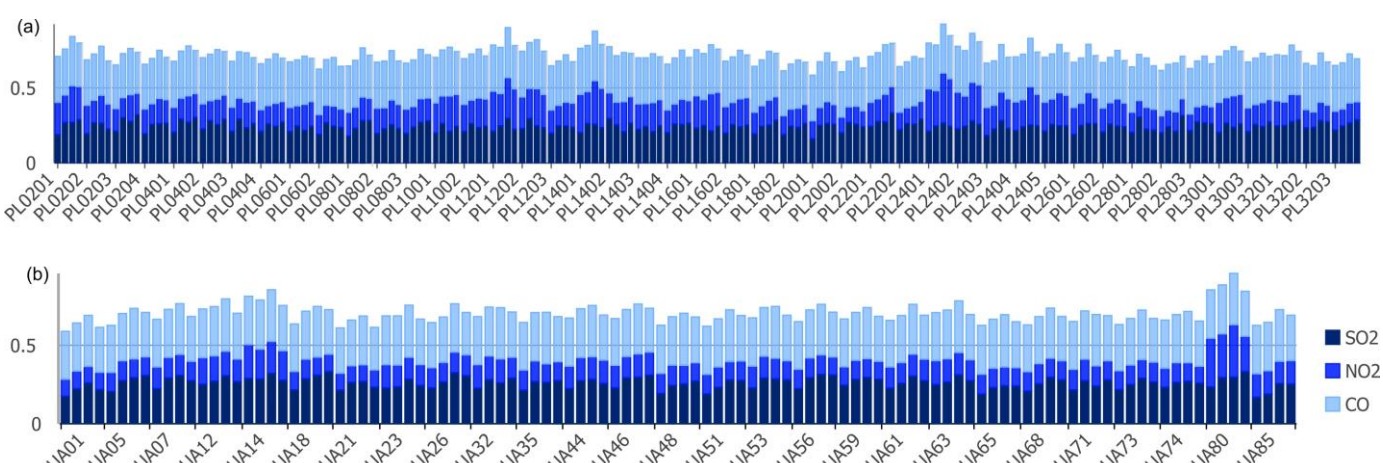

**Figure 14.** The sum of the normalized values of SO$_2$, NO$_2$, and CO gas concentrations for the years 2019, 2020, 2021, and 2022 from February 1 to October 31 in given areas; (**a**) air quality measurement zones in Poland and (**b**) administrative districts of Ukraine.

### 3.3.3. Spatiotemporal Variations

The percentage change in gas concentration between 2021 and 2022 (February to October) is shown in Figure 15. Appendix A includes a detailed statement of the percentage change of gas over the last three years, divided into zones.

In the analysis of SO$_2$ concentration for the whole study area, a decrease of approximately 25% in 2022 was recorded, in southern Poland by 18% and in eastern Ukraine and in the Crimean Peninsula by 16% (Figure 15a). On the other hand, an increase in SO$_2$ concentration by approximately 33% occurred on the Polish coast. Changes in 2021 were within ± 26%, with the largest increase of 26% in the port city of Sevastopol on the Crimean Peninsula and the highest decrease of 26% in the city of Olsztyn (PL2801). In 2020, most zones had an increase in SO$_2$ concentration of 15–30%. One of the SO$_2$ emitters is sea transport. It can be assumed that the increase in SO$_2$ concentration in 2021–2022 in the port areas results from the increased transport intensity.

The change in the level of NO$_2$ gas concentration in 2022 compared to the previous year ranges from −33% to 5% (Figure 15b). The greatest decrease in NO$_2$ concentration by 33% occurred in the city of Kyiv, a change that may have been caused by the outbreak of war and the emigration of the population. On the other hand, an increase of approximately 5% in NO$_2$ pollution occurred in Ukraine mostly in the Black Sea regions. Changes in 2021 are mostly an increase in concentration to 30%. The largest changes, almost 30%, occurred in the Silesian and Upper Silesian zones in southern Poland. In the same year, a maximum

decrease of 10% was recorded in the Kirovohradsk region in the central part of Ukraine. In 2020, 56 zones saw a decrease in $NO_2$ concentration to 21%. In 2020, a 1% increase in the concentration level was recorded only in the city of Elbląg (PL2802) in Poland. When analyzing the level of nitrogen dioxide concentration in 2021 in relation to the previous year and its certain increase, the impact of the situation related to the COVID-19 pandemic on the values recorded in 2020 should be taken into account. The restrictions introduced then by the lockdown resulted in a decrease in car traffic, including in cities, which are under the greatest pressure of pollution generated by car combustion engines, including nitrogen oxides.

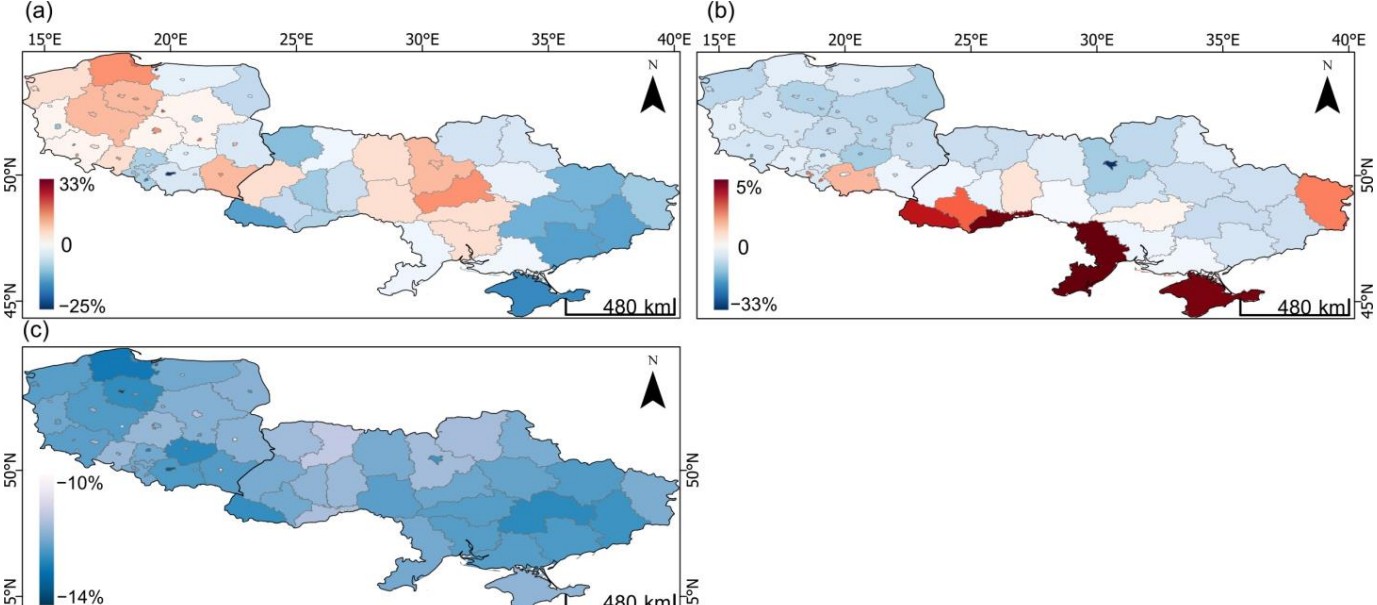

**Figure 15.** Percentage difference of $SO_2$, $NO_2$, and CO gas concentrations from 2021 to 2022 (February 1 to October 31); (**a**) percentage analysis of sulphur dioxide, (**b**) nitrogen dioxide, and (**c**) carbon monoxide.

Air pollution with CO gas in 2022 decreased by approximately 12–14% compared to the previous year (Figure 15c). In 2021, CO concentration increased by 5–7% in all regions. In 2020, the change in the level of CO concentration was within ± 2%. The increases in CO concentration in 2020 were observed in the north-eastern part of Ukraine and in the southern part of Poland. The measurement results show that such a large decrease in CO concentration in 2022 did not occur in the earlier period. There can be many reasons for changes; anthropogenic sources of CO, which account for 60% of emissions, are primarily mobile, industrial, and residential activities [42]. However, the introduction of legal norms regulates the levels of some pollution, e.g., for diesel passenger cars.

Then, trends in the speed of changes in the level of pollution in regions in Poland and Ukraine were identified. Full-time series of $SO_2$, $NO_2$, and CO data from 2018 to 2022 were analyzed. Linear regression and the non-parametric Mann–Kendall test for detecting the existence of a decreasing or increasing trend in the data series were used for statistical research.

Table 3 presents the results of statistical calculations regarding the trends in air pollution changes calculated from 4 years of satellite observations in selected zones in Poland and Ukraine. Zones of large cities in Poland and war zones in Ukraine were selected for presentation.

**Table 3.** A fragment of the test results for the regions showing trends of annual changes in gas concentrations in 2018–2022. Analyzed zones: Tri-City Agglomeration (PL2201), Donetska (UA14), Zaporizka (UA23), Warsaw Agglomeration (PL1401), Krakow Agglomeration (PL1201), and Kyivska (UA32).

| | Sulphur Dioxide | | | | Nitrogen Dioxide | | | | Carbon Monoxide | | | |
|---|---|---|---|---|---|---|---|---|---|---|---|---|
| Zone | Sen's Slope [μmol/m²] | *p* Value | L–R Slope [μmol/m²] | *p* Value | Sen's Slope [μmol/m²] | *p* Value | L–R Slope [μmol/m²] | *p* Value | Sen's Slope [μmol/m²] | *p* Value | L–R Slope [μmol/m²] | *p* Value |
| PL2201 | −32.96 | 0.12 | −109.60 | 0.13 | −0.73 | 0.17 | −1.15 | 0.23 | −507.86 | 0 | −554.30 | 0 |
| UA14 | −31.16 | 0.15 | −59.49 | 0.18 | −0.85 | 0.11 | −0.57 | 0.13 | −457.20 | 0 | −484.65 | 0.006 |
| UA23 | −21.61 | 0.17 | −32.54 | 0.18 | −0.53 | 0.18 | −0.50 | 0.24 | −451.20 | 0.0005 | −533.50 | 0.0007 |
| PL1401 | +4.79 | 0.51 * | +62.16 | 0.10 | −1.23 | 0.14 | −0.88 | 0.17 | −485.04 | 0 | −653.20 | 0 |
| PL1201 | −52.00 | 0 | −127.06 | 0 | −0.13 | 0.66 * | −0.34 | 0.49 * | −536.14 | 0 | −665.71 | 0 |
| UA32 | −22.18 | 0.18 | −44.27 | 0.15 | −1.18 | 0.05 | −1.12 | 0.12 | −431.69 | 0.0001 | −462.76 | 0.003 |

* The significance level of changes less than 75%.

Figure 16 shows the statistics of the rate of change in air pollution for each region from both areas. In the period 2018–2022, at the significance level of $p \in (0.10 \div 0.25)$, an increasing trend of air pollution with $SO_2$ gas was found in the Wroclaw Agglomeration and the zones of Vinnytska, Sevastopol, and Poltavska (Figure 16a). However, at the significance level of $p < 0.05$, an increasing trend was found in the city of Zielona Góra, and in other zones in Poland, a downward trend was observed. In central Ukraine, air quality is changing, with a particularly high decrease in $NO_2$ concentration recorded in Kyiv. An upward trend was demonstrated in the eastern part of Ukraine and in Poland in the zone of border crossings and in the west in the Lubuskie zone (Figure 16b). The result of the trend analysis for CO indicates a downward trend in air pollution with CO gas at the level of both countries (Figure 16c). Nevertheless, there are significant regional differences. The Podlaskie region has the largest downward trend (PL2002).

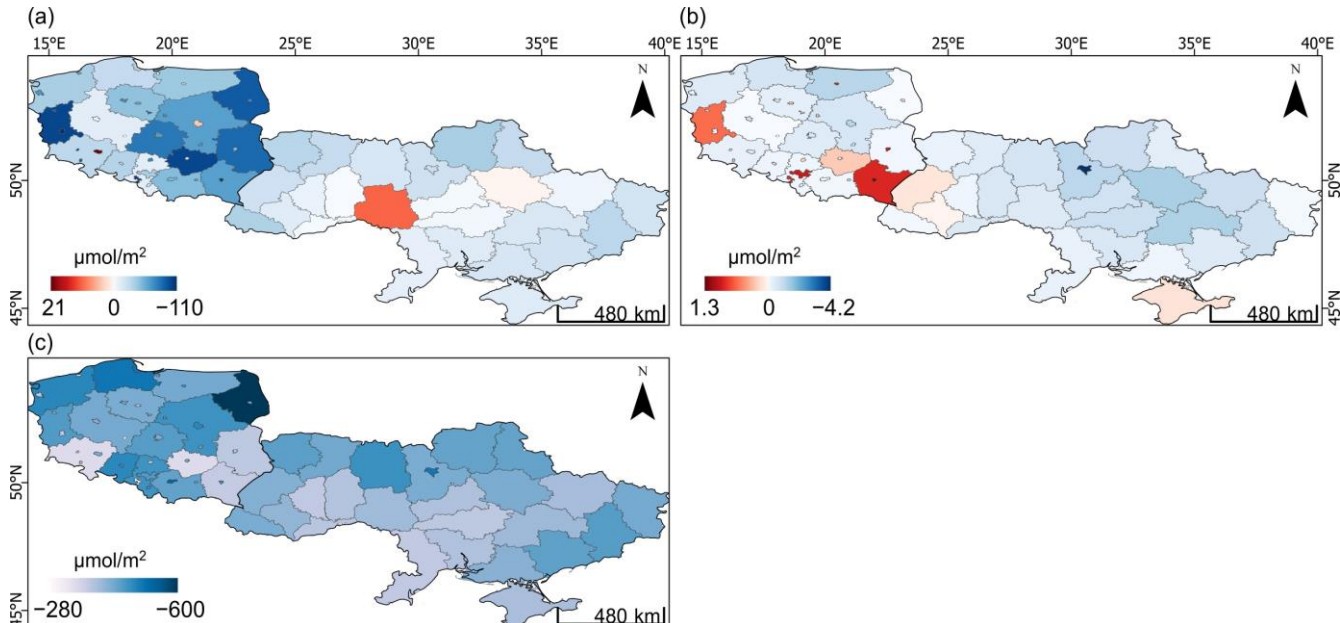

**Figure 16.** Trend results with non-parametric Mann–Kendall test of air pollution in zones in Poland and Ukraine; (**a**) $SO_2$, (**b**) $NO_2$, and (**c**) CO.

Additionally, indexes were calculated to indicate areas that were characterized by polluted air in 2022. They were calculated based on the following criteria: average gas concentration and the rate of change of $SO_2$, $NO_2$, and CO (Figure 17). In the study area, an index of 10 indicated the least polluted air, and an index of 7 indicated the air most polluted with $SO_2$, $NO_2$, and CO gases. When analyzing the results of air quality assessment in zones, it should be taken into account that the index of a specific zone is determined on

the basis of concentrations that occur in the entire area (including Moscow). The indices are obtained by analyzing the average measurement results from 2022 and the rate of change from November 2018 to October 2022. Based on the percentage composition of the air, carbon monoxide has the largest share, followed by nitrogen dioxide. Due to the fact that the average value of CO is the highest in the south of Poland, the lowest air quality index was obtained in the industrial Silesian zone. High indexes were calculated in large agglomerations such as Warsaw, Krakow, and Wroclaw due to the high content of air pollution with $NO_2$ gas. The Volynska zone in the north-west of Ukraine received the highest index.

### 3.3.4. The Impact of the War in Ukraine on Air Quality

Since 24 February 2022, military operations have been conducted in eastern, east-southern and northern parts of Ukraine (Figure 1a). Therefore, it was examined whether it was possible to designate significant points in the area of zones characterized by high air pollution in 2022. For this purpose, statistical tests (global Moran's I and Getis-Ord Gi*) were performed in ArcGIS Pro version 3.0.3.

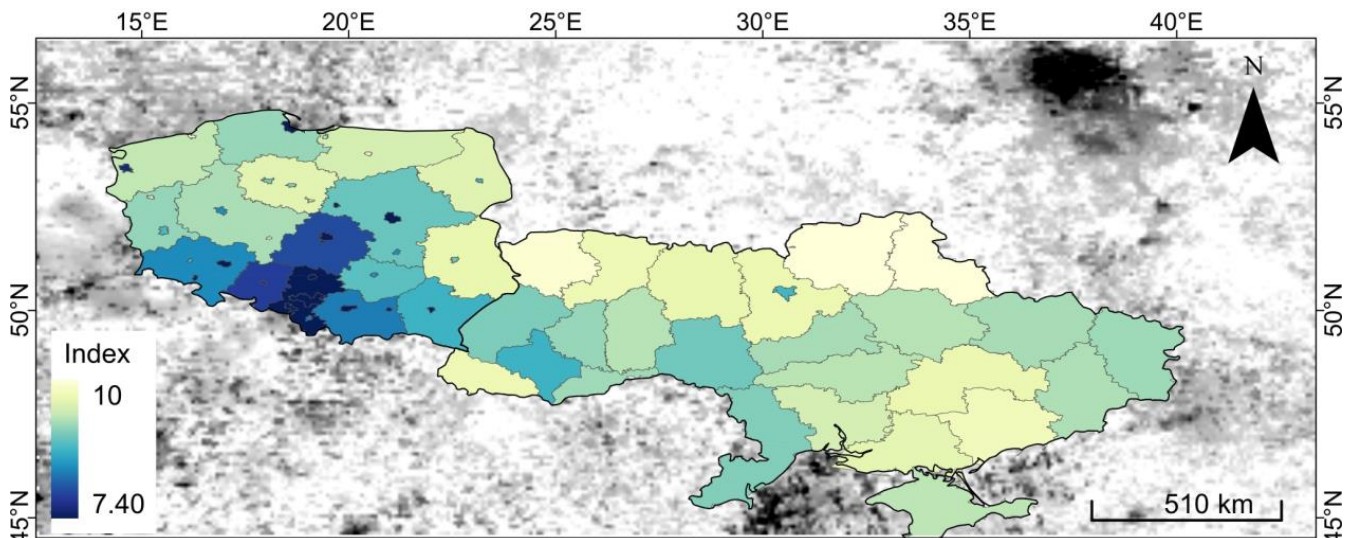

**Figure 17.** A map of spatial patterns of air quality in 2022.

Significant points were only assumed with a confidence level of 95% (Figure 18). For all analyzed $SO_2$ and CO data, there was a decrease in pollution (negative z-scores). On the other hand, the concentration of $NO_2$ gas increased from the beginning of 2022 in the Zakarpatska (UA21) zone and in the north in bordering zones, in the north of Kyiv, and in the Chernihivska (UA74) and Sumska (UA59) zones. The largest area is located in the Crimean Peninsula in the Autonomous Republic of Crimea zone (UA01). Based on percentage analyses of $SO_2$ concentration (Appendix A), this area showed decreasing changes. Air pollution with $SO_2$ gas in this area is due to warfare (https://ecodozor.org, accessed on 30 November 2022). The significant $NO_2$ points at a confidence level of 95% were observed primarily in the war zones in the south-eastern (UA14) and northern parts (UA74, UA59) of Ukraine. In zones to the north of Kyiv, areas with high concentrations of $NO_2$ tended to develop. When verifying these results with information from (https://ecodozor.org, accessed on 30 November 2022), it was found that many fires were recorded in these areas as a result of hostilities.

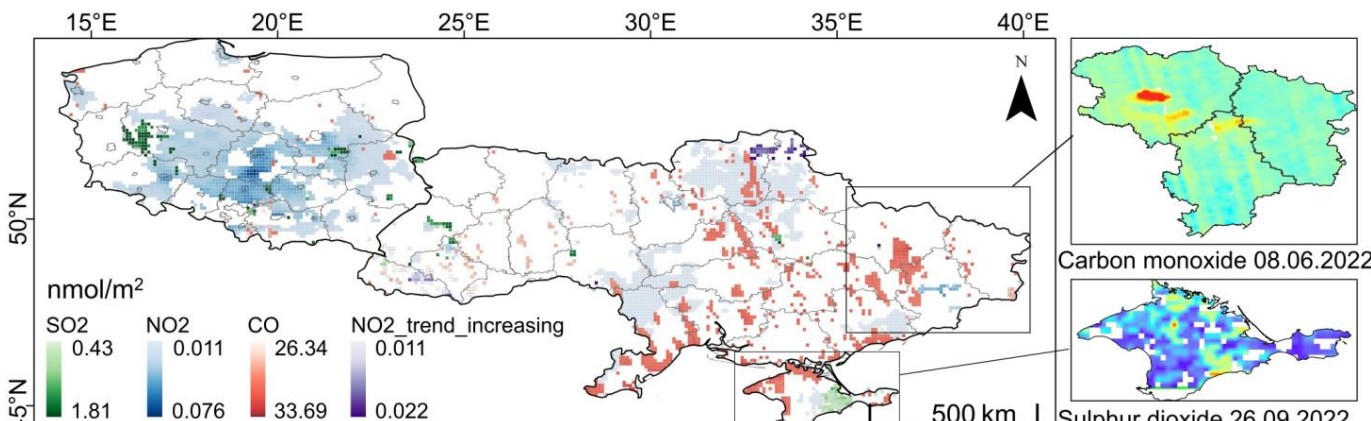

**Figure 18.** Analysis of significant points in Central and Eastern Europe during a 10-month period from 1 January 2022.

In the eastern and central part of Ukraine, an unusual spatial distribution of significant CO points was observed. Carbon Monoxide is the gas that persists longest in the air and can be transported over long distances [42]. There are many sources of CO emissions, both anthropogenic and natural, such as oxidation of biogenic hydrocarbons and methane oxidation.

In conclusion, in the territory of Ukraine, the pattern of the distribution of points is complex. The differences in $SO_2$, $NO_2$, and CO between model predictions and actual satellite measurements can be a proxy for the impact of armed conflict on atmospheric pollution. Figure 19 shows the difference between the forecasted concentration levels and the actual levels as the average values from February to October 2022. Gas concentration levels were forecast using the random forest-based adjusting method, the period from November 2018 to January 2022 was assumed with a monthly step.

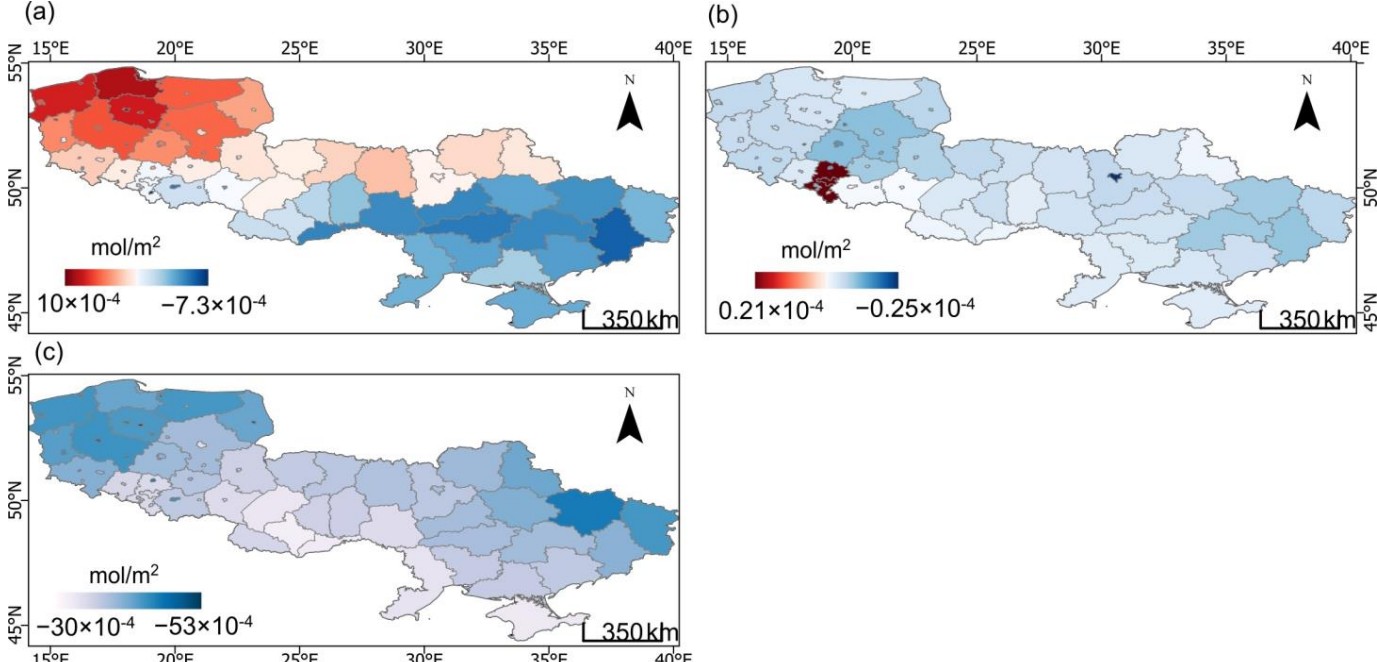

**Figure 19.** Difference in concentration levels between predicted and actual values during warfare: (**a**) $SO_2$, (**b**) $NO_2$, and (**c**) CO.

## 4. Discussion

TROPOMI-S5 satellite data are an important source in the field of spatiotemporal analyses of changes in the level of pollutant concentrations in the air. They complement existing ground-based monitoring methods but are often the primary source of air quality information, as in this case. Given the unlimited access to current data sets, it is possible to conduct continuous activities in the field of detecting the threat, measuring the degree of contamination, and assessing its effects. The concentration of air pollutants at a specific place and time depends on many factors, including emission sources, meteorology, landscape features, and physicochemical transformations. Certainly, in dynamic situations such as warfare, the main factors influencing the level of pollution are numerous anthropogenic emission sources. The range of data, the choice of interpretation pattern, and the amount of information, such as satellite imagery, are time-consuming. On the other hand, the connection with the topography or atmospheric circulation patterns requires an analysis of the impact of many factors of different scales. Som e types of pollutants show changes in emissions in the atmosphere in a clear and rapid way during the summer heat, whereas others do so during the winter. The persistence of gases in the atmosphere from several hours to a month [42] depends on the dispersion conditions, which are determined by various weather phenomena. A significant amount of information requires automation of processing, often using elements of artificial intelligence. Analysis of possibilities and limitations determined that the research area covered by this study would be the region of Central and Eastern Europe, including two neighboring countries: Poland and Ukraine. The general aim of this work was to determine the patterns of $SO_2$, $NO_2$, and CO air pollution in different time scales based on satellite measurements. For both countries, a zonal division was introduced, and for each zone spatial and temporal changes in the composition of the atmosphere were characterized, specifying gas concentrations. By introducing a regional analysis, it was possible to conclude how local zones interact with each other. Due to the time frame of this study (from 1 November 2018 to 31 October 2022), data from both before and during the ongoing military operations were collected. This study assessed the consequences of the invasion of the territory of Ukraine by the Russian army in relation to the quantitative levels of gas concentrations present in the atmosphere. The process of proceeding included sequential actions. Satellite data sets were pre-processed and generated, and could be used at the stage of calculating gas concentrations. The process involved evaluating the quality of the satellite dataset. Data from monitoring stations in Poland were taken as a reference. Then, statistical algorithms were used to calculate the temporal and spatial variability, allowing for the zonal assessment of air quality.

The rate of change for three pollution criteria was calculated in the paper. A downward trend was shown for $SO_2$ and CO in both Poland ($-161.67 \pm 5.48$ µmol/m$^2$, $-470.85 \pm 82.81$ µmol/m$^2$) and in Ukraine ($-32.56 \pm 23.51$ µmol/m$^2$, $-438.04 \pm 80.76$ µmol/m$^2$). On the other hand, for the concentration of $NO_2$ gas, an increasing trend was shown in Poland ($+0.67 \pm 0.47$ µmol/m$^2$), whereas a downward trend was shown in Ukraine ($-0.28 \pm 0.21$ µmol/m$^2$). One of the reasons for the declining trends is the use of policies as part of measures to improve air quality [62]. The main objectives are to reduce pollutant emissions at the source and to identify the most effective measures to reduce emissions at local and national levels. Another influence which affected the period under review were the restrictions related to the COVID-19 pandemic, when, among other things, a decrease in $NO_2$ concentration was noted. Currently, in Poland, the $NO_2$ concentration shows an upward trend.

Analyzing the zonal summary of distributions of average concentrations of $SO_2$, $NO_2$, and CO pollutants for each zone, from 2018 to 2022, one can clearly see the dependence of the level of concentrations ($SO_2$, $NO_2$, and CO) on the characteristics of a given zone. Individually, anthropogenic activity, energy production and distribution, industry, transport network, or agriculture can have a leading impact on the concentration level of a specific gas [62]. For example, the location of Ukraine's Zaporizhzhya Nuclear Power Plant in zone UA23 generates high average CO concentrations in all adjacent zones in the

south-eastern part of Ukraine (UA65, UA12, UA23). It can also be assumed that annual percentage changes in gas concentrations in the zones indicate local events. The city of Kyiv is a characteristic reference, for which a 33% decrease in $NO_2$ concentration was recorded in 2022. The authors in [5] estimated approximately 40% of the reduction in $NO_2$. An example of this is the increase in $SO_2$ concentration in 2022 on the Polish coast. This may be related to the increased sea transport of energy resources, which until now have been sent via a gas pipeline from Russia. Another case concerns a belt of several zones connecting both countries for which an increase trend in $NO_2$ concentration was observed. According to data from the Polish Border Guard, approximately 10 million refugees have crossed the Polish–Ukrainian border since 24 February 2022 (https://www.strazgraniczna.pl/, accessed on 20 February 2023). Mass migration of the population and the increasing movement of supplies by sea transport causes the release of additional pollutants into the air. Referring to the analyses, it can be assumed that the growth changes recorded in 2022 are related to the war in Ukraine. Lower concentrations were observed in the previous year; however, these cases should be considered in future studies. Estimation of all factors affecting air quality supported by additional analyses will provide a solution.

## 5. Conclusions

This study presents statistics that enable the general evaluation of changes in concentration levels of the gases $SO_2$, $NO_2$, and CO that had a major impact on air quality in the period 2018–2022. It was assessed that the average concentrations of $SO_2$ and CO gases are decreasing in Poland and Ukraine. On the other hand, in Poland, air pollution with $NO_2$ gas slightly increased ($+0.67 \pm 0.47$ μmol/m$^2$). The analysis included the division of the area into zones for each of the pollutants mentioned. Statistical calculations were the basis for the assessment of changes for individual pollutants. As a result, the zones where the air quality changes the fastest and the zones with the highest gas concentrations were defined. The obtained results indicate that the levels of air pollution in Central and Eastern Europe are characterized by significant variability both in terms of volume and the spatial differentiation of their occurrence.

Analyses of percentage changes and the use of the sequential Mann–Kendall test to investigate the 4-year (2018–2022) trends of $SO_2$, $NO_2$, and CO from these source areas confirmed the variability and zonal dependence of air pollution. The effect of war on the level of air pollution by various gases is a topic that requires further research. It will be important to determine the share of individual emitters. Therefore, further work will be related to the development of a matrix of influence of meteorological and topographic factors and emitters to verify the zone space-time characteristics presented in this work.

**Funding:** This research received no external funding.

**Data Availability Statement:** Sentinel-5P data are freely available from the Goddard Earth Sciences Data and Information Services Center (GES DISC). Observations from measurement stations are freely available from the Polish Chief Inspectorate of Environment in Poland.

**Conflicts of Interest:** The author declares no conflict of interest.

## Appendix A

**Table A1.** The list of zone names and their code; trend direction and percentage change.

| Code | Area Name | Trend $SO_2$ 2018–2022 | Change $SO_2$ [%] | | | Trend $NO_2$ 2018–2022 | Change $NO_2$ [%] | | | Trend CO 2018–2022 | Change CO [%] | | |
|---|---|---|---|---|---|---|---|---|---|---|---|---|---|
| | | | 2019–2020 | 2020–2021 | 2021–2022 | | 2019–2020 | 2020–2021 | 2021–2022 | | 2019–2020 | 2020–2021 | 2021–2022 |
| PL0201 | Wroclaw Agglomeration | ↑ | 30.0 | 0.4 | 5.5 | ↓ | −15.1 | 24.9 | −8.8 | ↓ | 0.04 | 6.8 | −11.7 |
| PL0202 | city Legnica | ↓ | 26.9 | −1.4 | −14.4 | ↓ | −20.7 | 19.9 | −11.0 | ↓ | 1.6 | 5.7 | −12.2 |
| PL0203 | city Wałbrzych | ↓ | 29.6 | −8.1 | 13.5 | ↑ | −10.4 | 24.7 | −19.1 | ↓ | 0.7 | 4.8 | −11.5 |

**Table A1.** *Cont.*

| Code | Area Name | Trend SO$_2$ 2018–2022 | Change SO$_2$ [%] | | | Trend NO$_2$ 2018–2022 | Change NO$_2$ [%] | | | Trend CO 2018–2022 | Change CO [%] | | |
|---|---|---|---|---|---|---|---|---|---|---|---|---|---|
| | | | 2019–2020 | 2020–2021 | 2021–2022 | | 2019–2020 | 2020–2021 | 2021–2022 | | 2019–2020 | 2020–2021 | 2021–2022 |
| PL0204 | zone Lower Silesian | ↓ | 24.3 | 2.3 | 0.5 | ↓ | −16.5 | 17.6 | −5.7 | ↓ | 0.9 | 6.0 | −12.3 |
| PL0401 | Bydgoska Agglomeration | ↓ | 29.9 | −7.1 | 10.7 | ↓ | −15.8 | 20.9 | −9.9 | ↓ | 1.8 | 5.6 | −13.7 |
| PL0402 | city Toruń | ↓ | 19.6 | −9.4 | 11.8 | ↑ | −20.8 | 21.8 | −4.5 | ↓ | 0.7 | 6.2 | −12.1 |
| PL0403 | city Włocławek | ↓ | 27.3 | −19.8 | 10.9 | ↑ | −12.4 | 18.6 | −12.3 | ↓ | 1.0 | 5.9 | −12.3 |
| PL0404 | zone Kujawsko-pomorska | ↓ | 19.1 | −6.3 | 10.6 | ↓ | −15.1 | 20.3 | −10.7 | ↓ | 0.9 | 6.3 | −12.5 |
| PL0601 | Lubelska Agglomeration | ↓ | 15.3 | −12.0 | 11.2 | ↑ | −16.5 | 22.9 | −5.0 | ↓ | 0.4 | 4.7 | −10.5 |
| PL0602 | zone Lubelska | ↓ | 29.0 | −8.5 | −3.6 | ↓ | −10.8 | 12.4 | −8.3 | ↓ | 0.1 | 6.2 | −11.9 |
| PL0801 | city Gorzów Wlkp. | ↓ | 22.3 | 18.5 | −0.2 | ↓ | −10.7 | 9.7 | −5.3 | ↓ | 1.3 | 4.5 | −12 |
| PL0802 | city Zielona Góra | ↓ | 13.8 | 10.6 | −9.8 | ↓ | −17.2 | 15.1 | −2.8 | ↓ | 1.6 | 5.3 | −12.2 |
| PL0803 | zone Lubuska | ↓ | 17.3 | 11.7 | 3.3 | ↑ | −14.9 | 13.0 | −3.4 | ↓ | 1.2 | 5.4 | −12.1 |
| PL1001 | Agglomeration Łódzka | ↓ | 23.8 | −19.5 | 13.5 | ↓ | −7.7 | 21.5 | −9.6 | ↓ | 0.8 | 5.8 | −12 |
| PL1002 | zone Łódzka | ↓ | 20.0 | −9.9 | 3.9 | ↓ | −12.7 | 20.3 | −8.4 | ↓ | 0.5 | 5.7 | −11.8 |
| PL1201 | KrakowAgglomeration | ↓ | 14.3 | 16.9 | −24.5 | ↓ | −19.5 | 20.8 | −0.5 | ↓ | 0.1 | 7.4 | −13.4 |
| PL1202 | city Tarnów | ↓ | 22.1 | −16.1 | −5.1 | ↓ | −4.1 | 18.6 | −10.4 | ↓ | 0.7 | 6.1 | −11.9 |
| PL1203 | zone Małopolska | ↓ | 19.6 | 0.4 | −3.7 | ↓ | −12.1 | 14.1 | 1.5 | ↓ | 0.3 | 6.3 | −12.4 |
| PL1401 | Warsaw Agglomeration | ↑ | 22.9 | −1.8 | −8.6 | ↓ | −16.4 | 27.0 | −10.0 | ↓ | 0.3 | 6.7 | −11.5 |
| PL1402 | city Płock | ↓ | −13.9 | −18.2 | 21.5 | ↑ | −11.8 | 24.9 | −12.1 | ↓ | 0.5 | 5.4 | −11.8 |
| PL1403 | city Radom | ↓ | 7.3 | −13.7 | 15.9 | ↓ | −9.4 | 20.5 | −10.3 | ↓ | −0.6 | 6.3 | −12.2 |
| PL1404 | zone Mazowiecka | ↓ | 22.0 | −1.0 | 2.6 | ↓ | −11 | 20.2 | −9.9 | ↓ | 0.2 | 6.3 | −12 |
| PL1601 | city Opole | ↓ | 7.2 | −14.7 | 12.9 | ↓ | −19.7 | 30.2 | −8.9 | ↓ | −0.02 | 6.0 | −11.9 |
| PL1602 | zone Opolska | ↓ | 21.8 | −5.2 | 4.9 | ↓ | −16.2 | 20.7 | −3.5 | ↓ | 0.1 | 6.2 | −11.8 |
| PL1801 | City Rzeszów | ↓ | 20.3 | 3.5 | 11.3 | ↑ | −4.2 | 16.8 | −5.7 | ↓ | −0.2 | 6.0 | −11 |
| PL1802 | zone Podkarpacka | ↓ | 22.9 | −3.7 | 10.3 | ↑ | −9.9 | 12.8 | −2.0 | ↓ | 0.3 | 6.2 | −12.3 |
| PL2001 | Białostocka Agglomeration | ↓ | 35.5 | 4.8 | −1.9 | ↑ | −1.0 | 18.4 | −10.3 | ↓ | 0.7 | 5.8 | −12.3 |
| PL2002 | zone Podlaska | ↓ | 25.4 | −4.5 | −6.5 | ↓ | −0.5 | 13.4 | −11.3 | ↓ | 0.5 | 6.3 | −11.9 |
| PL2201 | Tri-City Agglomeration | ↓ | 11.4 | −0.7 | 17.7 | ↓ | −5.1 | 17.9 | −4.4 | ↓ | 0.5 | 6.8 | −12.5 |
| PL2202 | zone Pomorska | ↓ | 15.7 | −1.9 | 11.3 | ↓ | −10.2 | 15.6 | −3.5 | ↓ | 0.5 | 6.6 | −12.8 |
| PL2401 | Upper Silesian Agglomeration | ↓ | 13.7 | 7.3 | −7.9 | ↑ | −16.1 | 29.3 | −4.9 | ↓ | −0.4 | 6.7 | −12.1 |
| PL2402 | Rybnicko-Jastrzębska Agglomeration | ↓ | 6.8 | 12.9 | −8.5 | ↑ | −17.7 | 21.6 | 2.5 | ↓ | −0.7 | 7.1 | −12.1 |
| PL2403 | city Bielsko-Biała | ↓ | 18.6 | 19.9 | −18.1 | ↓ | −14.2 | 16.5 | 2.8 | ↓ | −0.2 | 6.6 | −12.2 |
| PL2404 | city Częstochowa | ↓ | 10.2 | 5.3 | −2.4 | ↑ | −5.0 | 28.6 | −17.5 | ↓ | 0.8 | 5.9 | −12.7 |
| PL2405 | zone Silesia | ↓ | 13.7 | 7.3 | −7.9 | ↓ | −16.1 | 29.3 | −4.9 | ↓ | −0.4 | 6.7 | −12.1 |
| PL2601 | city Kielce | ↓ | 24.2 | 4.9 | −0.6 | ↓ | −17.9 | 28.1 | −16.5 | ↓ | 0.05 | 6.3 | −12.3 |
| PL2602 | zone Świętokrzyska | ↓ | 20.0 | −4.8 | −2.0 | ↑ | −10.6 | 20.5 | −11.8 | ↓ | −0.3 | 6.7 | −12.6 |
| PL2801 | city Olsztyn | ↓ | 32.9 | −26.1 | −2.9 | ↑ | −16.7 | 23.9 | −5.3 | ↓ | 1.4 | 6.6 | −11.6 |
| PL2802 | city Elbląg | ↓ | 16.2 | −12.4 | 32.9 | ↓ | 0.6 | 13.6 | −12.4 | ↓ | 0.4 | 6.4 | −13 |
| PL2803 | zone Warmińsko-mazurska | ↓ | 20.9 | −2.1 | −2.1 | ↓ | −8.0 | 15.0 | −5.5 | ↓ | 0.6 | 6.3 | −12.1 |
| PL3001 | Poznańska Agglomeration | ↓ | 23.1 | −10.0 | 8.3 | ↓ | −15.2 | 19 | −6.2 | ↓ | 2.0 | 4.4 | −11.9 |
| PL3003 | zone Wielkopolska | ↓ | 18.4 | −4.9 | 11.2 | ↔ | −15.7 | 15.6 | −6.1 | ↓ | 1.2 | 5.7 | −12.3 |
| PL3201 | Szczecińska Agglomeration | ↓ | 0.8 | 9.3 | 4.4 | ↓ | −6.1 | 13.9 | −7.6 | ↓ | 0.8 | 5.7 | −12.3 |
| PL3202 | city Koszalin | ↓ | 1.8 | 15.7 | −3.0 | ↓ | −17.1 | 17.1 | −9.1 | ↓ | 0.4 | 5.6 | −12.1 |
| PL3203 | zone Zachodniopomorska | ↓ | 11.7 | 7.3 | 7.4 | ↓ | −10.9 | 15.8 | −9.5 | ↓ | 0.8 | 5.7 | −12.3 |
| UA01 | Autonomous Republic of Crimea | ↓ | 21.3 | 14.6 | −16.3 | ↑ | 3.4 | −7.2 | 4.7 | ↓ | 0.5 | 5.8 | −11.9 |
| UA05 | Vinnytska | ↑ | 25.6 | 6.3 | 4.5 | ↓ | 3.1 | −5.1 | −0.6 | ↓ | 0.3 | 6.5 | −12.3 |
| UA07 | Volynska | ↓ | 23.2 | 5.4 | −10.6 | ↓ | −4.9 | 1.5 | −8.0 | ↓ | 0.4 | 5.5 | −11.7 |
| UA12 | Dnipropetrovska | ↓ | 7.2 | 11.2 | −12.0 | ↓ | −4.8 | −3.1 | −8.1 | ↓ | 1.1 | 5.9 | −12.6 |
| UA14 | Donetska | ↓ | −1.4 | 11.2 | −13.7 | ↓ | −11.7 | 6.5 | −7.6 | ↓ | 1.6 | 5.0 | −12.5 |
| UA18 | Zhytomyrska | ↓ | 24.6 | 7.4 | 6.9 | ↓ | 5.4 | −7.1 | −3.7 | ↓ | 1.2 | 5.0 | −12.1 |
| UA21 | Zakarpatska | ↓ | 18.8 | 2.5 | −13.8 | ↑ | −1.4 | −0.2 | 4.0 | ↓ | 0.4 | 6.3 | −12.5 |
| UA23 | Zaporizka | ↓ | 3.2 | 16.8 | −14.0 | ↓ | −7.4 | 1.9 | −5.8 | ↓ | 0.9 | 5.4 | −12.3 |
| UA26 | Ivano-Frankivska | ↓ | 14.5 | 18.1 | −6.3 | ↑ | −3.5 | 4.8 | 2.7 | ↓ | 0.5 | 6 | −12.1 |
| UA32 | Kyivska | ↓ | 19.3 | −7.0 | 10.0 | ↓ | 0.4 | 0.8 | −12.1 | ↓ | 1.8 | 4.4 | −11.7 |
| UA35 | Kirovohradska | ↓ | 20.6 | −1.9 | 4.5 | ↓ | 3.0 | −10.5 | 0.3 | ↓ | 1.4 | 6.1 | −12.3 |
| UA44 | Luhanska | ↓ | 18.5 | 2.8 | −9.1 | ↓ | 3.1 | −1.8 | 2.3 | ↓ | 1.6 | 4.7 | −12.0 |
| UA46 | Lvivska | ↓ | 21.9 | 1.2 | 4.3 | ↑ | −6.0 | 9.1 | −1.6 | ↓ | 0.6 | 5.8 | −12.1 |
| UA48 | Mykolaivska | ↓ | 21.2 | 3.4 | 6.8 | ↓ | 0.1 | −6.9 | −1.9 | ↓ | 0.6 | 6.8 | −12.3 |
| UA51 | Odeska | ↓ | 18.1 | 16.8 | −1.0 | ↓ | 3.4 | −9.3 | 4.9 | ↓ | 0.1 | 6.9 | −12.1 |
| UA53 | Poltavska | ↑ | 21.7 | −1.9 | −1.9 | ↓ | −0.3 | −5.1 | −7.5 | ↓ | 0.8 | 6.0 | −12.2 |
| UA56 | Rivnenska | ↓ | 23.7 | 6.6 | −1.0 | ↓ | 1.8 | −1.3 | −7.7 | ↓ | 0.7 | 5.1 | −11.4 |
| UA59 | Sumska | ↓ | 13.3 | 4.0 | −4.0 | ↑ | 2.7 | −0.6 | −4.1 | ↓ | −0.004 | 6.2 | −12.0 |
| UA61 | Ternopilska | ↓ | 11.9 | 15.0 | −9.2 | ↓ | −1.9 | .5.0 | −2.4 | ↓ | 0.9 | 5.7 | −11.9 |
| UA63 | Kharkivska | ↓ | 6.2 | 14.6 | −11.7 | ↓ | −3.5 | −3.6 | −7.3 | ↓ | 1.6 | 5.5 | −12.3 |

**Table A1.** *Cont.*

| Code | Area Name | Trend SO₂ 2018–2022 | Change SO$_2$ [%] | | | Trend NO₂ 2018–2022 | Change NO$_2$ [%] | | | Trend CO 2018–2022 | Change CO [%] | | |
|---|---|---|---|---|---|---|---|---|---|---|---|---|---|
| | | | 2019–2020 | 2020–2021 | 2021–2022 | | 2019–2020 | 2020–2021 | 2021–2022 | | 2019–2020 | 2020–2021 | 2021–2022 |
| UA65 | Khersonska | ↓ | 18.4 | 4.8 | −1.0 | ↓ | −2.8 | −3.1 | −1.6 | ↓ | 0.1 | 6.4 | −12.3 |
| UA68 | Khmelnytska | ↓ | 18.3 | 14.0 | −5.6 | ↓ | 4.3 | −5.4 | 0.8 | ↓ | 0.6 | 6.1 | −11.8 |
| UA71 | Cherkaska | ↓ | 21.2 | −12.6 | 14.3 | ↓ | 4.1 | −5.3 | −5.6 | ↓ | 1.1 | 5.9 | −12.1 |
| UA73 | Chernivetska | ↓ | 13.4 | 13.2 | −7.8 | ↓ | 2.3 | −2.5 | 4.8 | ↓ | 0.9 | 5.8 | −11.6 |
| UA74 | Chernihivska | ↓ | 11.8 | 2.5 | −4.5 | ↓ | 3.1 | −5.6 | −9.1 | ↓ | 0.2 | 6.0 | −11.7 |
| UA80 | Kyiv | ↓ | 19.9 | 1.2 | 10.7 | ↓ | −10.0 | 16.5 | −32.9 | ↓ | 1.3 | 4.7 | −12.5 |
| UA85 | Sevastopol | ↑ | 10.7 | 26.0 | −0.9 | ↓ | −0.9 | −4.3 | 4.7 | ↓ | −0.02 | 6.2 | −12.1 |

↑—An increasing trend, ↓—a decreasing trend.

## Appendix B

Figure A1 Comparison of satellite measurements with terrestrial measurements for stations, at which the level of permissible levels of sulfur dioxide in the air was noted.

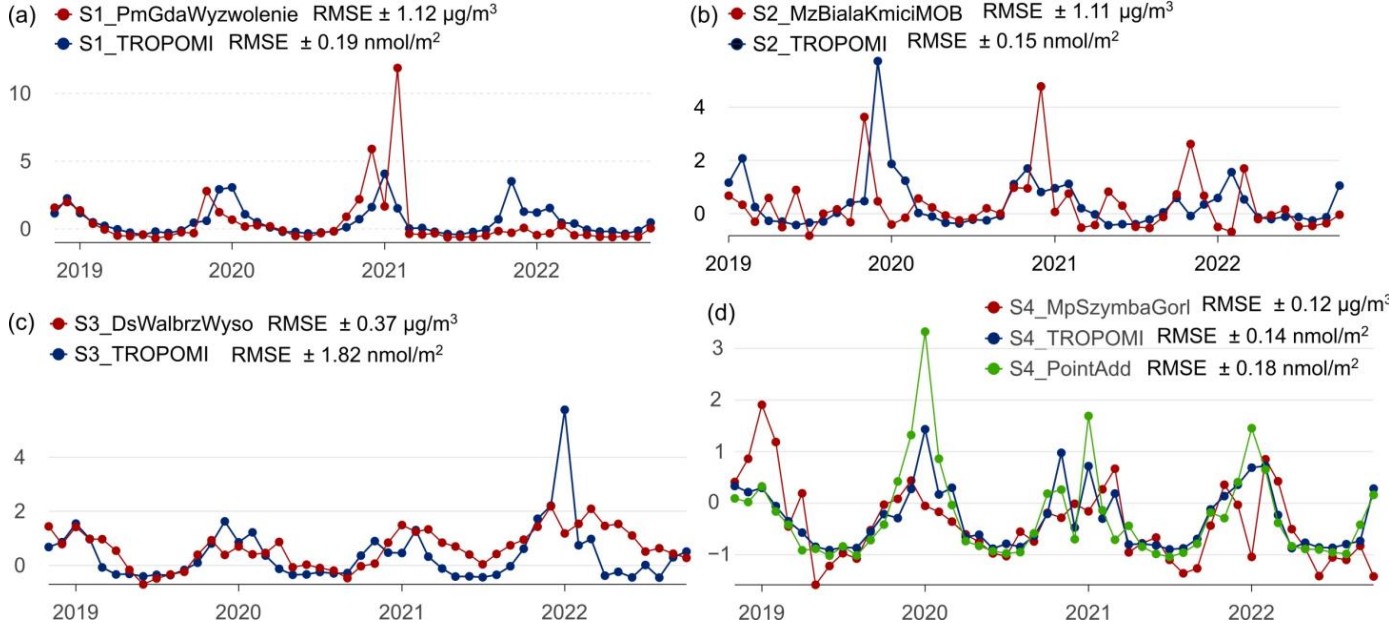

**Figure A1.** Time series of average monthly measurements of SO2 concentration from ground-based monitoring stations and satellite observations at the location of the stations in 2018–2022; (**a**) station in Gdansk (PmGdaWyzwolenie *r* = 0.38); (**b**) station in Płock (MzBialaKmiciMOB *r* = 0.48); (**c**) the station in Wałbrzych (DsWalbrzWyso *r* = 0.44); and (**d**) the station in Szymbark (MpSzymbaGorl *r* = 0.55). Values of r-Pearson correlation coefficient for stations at *p* < 0.05.

Figure A2 Comparison of satellite measurements with terrestrial measurements for selected stations, which recorded exceedances of the level of permissible nitrogen dioxide concentration in the air.

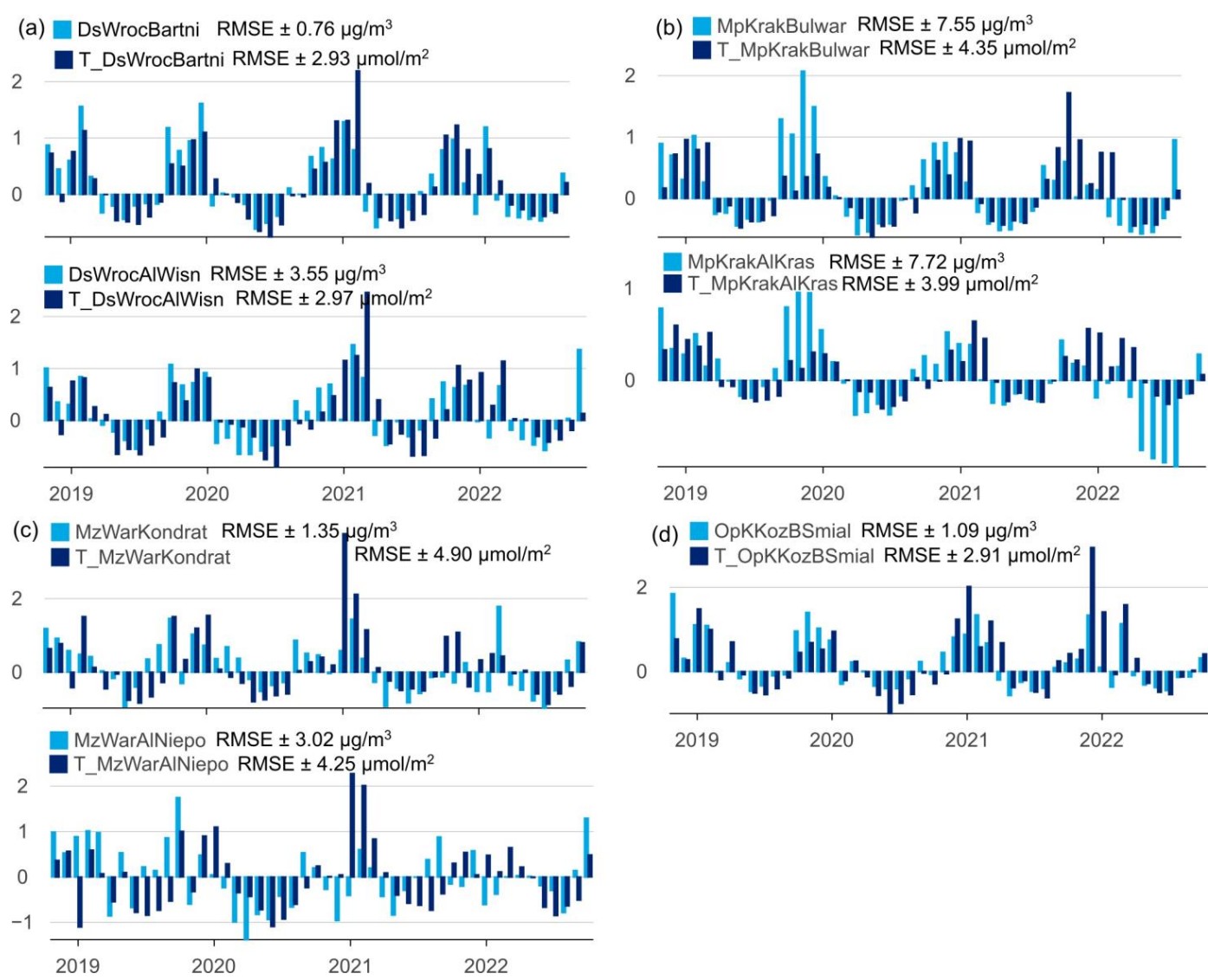

**Figure A2.** Selected time series of average monthly measurements of NO$_2$ concentration from ground-based monitoring stations and satellite observations at the location of the stations in 2018–2022; (**a**) stations in Wroclaw (DsWrocBartni $r = 0.83$, DsWrocAlWisn $r = 0.67$); (**b**) stations in Kraków (MpKrakBulwar, MpKrakAlKras $r \in (0.58, 0.61)$); (**c**) stations in Warsaw (MzWarKondrat $r = 0.55$, in the city center MzWarAlNiepo $r = 0.32$); and (**d**) station in Kedzierzyn-Kozle (OpKKozBSmial $r = 0.77$). Values of r-Pearson correlation coefficient for stations at $p < 0.05$.

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
