# Peer review of "Air Pollution Patterns Mapping of SO2, NO2, and CO Derived from TROPOMI over Central-East Europe"

_remotesensing, doi:10.3390/rs15061565_

Round 1

Reviewer 1 Report

This paper investigates the levels and patterns of air pollutant concentrations (CO, NO2, SO2) in Polland and Ukraine in the period 2018-2022. It compares satellite (TROPOMI)and ground station measurements in Poland (not in Ukraine) and presents maps of spatial change in percentage from 2021 to 2022 for both Poland and Ukraine. The idea is interesting and the analysis is well-performed and scientifically correct. However, the paper is too long and needs to be simplified. There are several parts that could be omitted in order to make it easier to follow, for instance, the validation of the satellite measurements with ground data could be placed in the appendix and the analysis could be separated into "temporal changes" and "spatial changes". The connection between the war and the specific pollutants considered is not well described and needs to be developed (what is the expected variation on CO, NO2, SO2 in a war zone? Are there other studies in war zones reporting similar variations?). Also, it would be interesting if the author could connect the results with atmospheric circulation patterns. The English needs to be improved as well.

Specific comments:

-Figure 1- the texture (pattern) of NO2 is not visible anywhere

-Reference [41] and lines onward. It is not clear if you are describing your study or the one referred

-Formulas and indexes should not be presented in the datasets (line 179)

-typos: air pollutio (line 314), interes (line 266)

-paragraph 304-313 out of place (move to above)

-The description of the results should be more focussed on what really matters and not just an enumeration of what pictures display

-The discussion needs to be separated from the results and needs to be better developed. Why are the trends decreasing? What could be the reason?

Author Response

Dear Reviewer,

I would like to thank you for your constructive comments, which helped to improve my manuscript „Air pollution patterns mapping of SO2, NO2, and CO derived from TROPOMI over Central-East Europe”. Below you will find the answers to the comments one by one.

Thank you for your time and effort.

Sincerely,

 Beata Wieczorek

Reviewer 2 Report

Author estimates the amount of pollutant concentrations and their patterns of spatio-temporal change in Central and Eastern Europe based on TROPOMI and ground observations. This is reasonable research about air pollution considering international event, and author conducts much work. Based on careful review, I could offer some suggestions to improve this manuscript. And a better version is expected.

1. There are some grammatical and clerical errors, and it’s necessary to carefully check and revise this manuscript. Furthermore, many parentheses with words are confusing.

2. Abstract should be condensed.

3. Keywords need to be improved, and it’s not necessary to list all pollutants.

4. Introduction needs to be refined, and the relationship with previous studies and the motivation of this work should be clarified.

5. The novelty and significance of this work should be further highlighted in manuscript.

Author Response

(The authors gave the same response as above.)

Reviewer 3 Report

The manuscript “Air pollution patterns mapping of SO2, NO2, and CO derived from TROPOMI over Central-East Europe” is showing trace gasses patterns over Central-East Europe from 2018 to 2022. The manuscript is well-written, however, the author claimed that the increase of NO2 in 2022 is mainly due to the military operation in Ukraine which is not supported by results and data analysis because higher concentrations of NO2 were also observed in the previous year. Further analyses are required to support the results, especially the link of increased NO2 with military operations. 

Author Response

(The authors gave the same response as above.)

Round 2

Reviewer 1 Report

The authors did a good job and addressed all the comments I previously made.

Reviewer 2 Report

Author has made  improvements according to my suggestions, and I have no further comments. Editor could give a consideration to this manuscript  based on comments of all reviewers. 

Reviewer 3 Report

Thanks to the author for revising the manuscript and justify the reviewer's comment. No further comments.